# Lead binds HIF-1α contributing to depression-like behaviour through modulating mitochondria-associated astrocyte ferroptosis

Fan Shi [1,2,7], Yuran Wang[1,2,7], Han Hao[1,2], Yuwei Zhao[1,2], Fanwu Wu[3], Yanlei Ge[4], Shun-Cheng Liu[5], Pengcai Liu[5], Weixuan Wang [1,2] ✉ & Yanshu Zhang [1,2,6] ✉

Environmental lead (Pb) exposure has been implicated in the development of depression. Here we show that Pb induces depression-like behaviour in mice by triggering ferroptosis. Single-cell RNA sequencing revealed that astrocytes exhibited the highest ferroptosis scores in brain tissue following Pb exposure. Further analysis identified HIF-1α as an early regulator of ferroptosis-related genes in astrocytes. Computational model and experimental validation demonstrated that $Pb^{2+}$ binds to the proline hydroxylation site of HIF-1α at position 138, stabilizing the protein structure and facilitating its nuclear translocation. Subsequently, the ferroptosis-related DEGs of the transcriptional regulatory network of HIF-1α in astrocyte ferroptosis were mainly enriched in mitochondrial dysfunction, wherein the role of voltage-dependent anion channel 1 (VDAC1) is notable. Further experiments also confirmed that HIF-1α promotes VDAC1 transcription, regulating mitochondrial dysfunction in astrocytes following Pb exposure. These findings indicate that astrocyte ferroptosis contributes to depression-like behaviour following Pb exposure through the HIF-1α/VDAC1 axis. Pb appears to stabilize HIF-1α by binding to hydroxylation site, preventing its degradation and promoting downstream mitochondrial dysfunction.

Worldwide, the incidence of depression is increasing annually, resulting in greater economic and social burdens[1]. Environmental factors, particularly long-term low-dose lead (Pb) exposure, are recognized as one of the key contributing factors[2,3]. However, the underlying mechanisms remain unclear. Ferroptosis, a form of programmed cell death, is characterized by $Fe^{2+}$ accumulation, lipid peroxidation, and GSH depletion. Additionally, it has been implicated in the pathophysiology of depression[4,5]. The biological processes of ferroptosis are significantly activated during chronic social defeat stress and alcohol-induced depression-like behaviour in mice[6,7]. However, the role of ferroptosis in Pb exposure-induced depression-like behaviours is unclear. Different cell types exhibit varying sensitivities to ferroptosis under diverse stimuli due to differences in their capacities to store iron and inhibit lipid peroxidation[8,9]. Primary microglia, astrocytes, and neurons demonstrate differential susceptibility to ferroptosis upon RSL3 treatment[10]. However, the effect of Pb exposure on the sensitive cell type in the mouse brain remains unclear.

Early identification of key ferroptosis-related genes may help slow or prevent disease progression[11,12]. Promoting ferroptosis in hepatic stellate cells has been shown to counteract liver fibrosis, with ALOX15 identified as a critical ferroptosis driver gene in these cells[13]. Additionally, conjugated fatty acids induce ferroptosis by targeting mitochondria and promoting chaperone-mediated autophagic degradation of GPX4, demonstrating potent antitumor effects[14]. However, it is unclear that the critical gene of ferroptosis driver following Pb exposure in depression-like behaviours.

[1]School of Public Health, North China University of Science of Technology, Tangshan, Hebei, China. [2]Hebei Key Laboratory of Occupational Health and Safety for Coal Industry, Tangshan, Hebei, China. [3]Traditional Chinese Medical College, North China University of Science of Technology, Tangshan, Hebei, China. [4]Department of Pulmonary and Critical Care Medicine, North China University of Science and Technology Affiliated Hospital, Tangshan, Hebei, China. [5]Hebei Key Laboratory of Medical Engineering and Integrated Utilization of Saline alkali Land, North China University of Science and Technology, Tangshan, Hebei, China. [6]School of Life Sciences, North China University of Science and Technology, Tangshan, Hebei, China. [7]These authors contributed equally: Fan Shi, Yuran Wang. ✉e-mail: wangwx@ncst.edu.cn; yanshuzhang@ncst.edu.cn

Mitochondrial dysfunction is increasingly recognized as a key contributor to the initiation and progression of ferroptosis[15]. Mitochondria tightly regulate iron homeostasis and ROS production in normal condition[16]. Under pathological conditions or environmental stress, excess iron and reactive oxygen species (ROS) can leak from mitochondria into the cytosol through mitochondrial membrane channels promoting the Fenton reactions and process of ferroptosis[17,18]. Studies have shown that sodium palmitate induces ferroptosis in MIN6 cells via PINK1/Parkin-Mediated Mitophagy[19]. Cadmium exposure triggers ferroptosis in hippocampal neurons through the mtROS-ferritinophagy axis, highlighting the important role of mitochondria in ferroptosis[20]. However, the role of mitochondria in cell ferroptosis induced by Pb exposure remains unclear.

Herein, we established a Pb-exposed mouse model to explore whether ferroptosis is involved in depression-like behaviour following Pb exposure. ScRNA-seq and biochemistry analyses were used to identify the most sensitive cells and perturbate early gene to ferroptosis in mouse brain tissue following Pb exposure. Our findings provide new insights into the mechanisms underlying Pb-induced depression-like behaviour.

## Results

### Ferroptosis is involved in depression-like behaviors of mice following Pb exposure

Behavioural tests were conducted including the forced swim (FST), tail suspension (TST), elevated plus maze (EPM), open field (OFT), and sucrose preference tests (SPT) to assess depression-like behaviours of mice following Pb exposure (Fig. 1A). The results indicated that both the time spent in the central squares and the overall distance travelled in the OFT were reduced in the MPb and HPb groups compared to those in the control group (Fig. 1B–D). Similarly, mice in the LPb, MPb, and HPb groups spent less time in the open arm, demonstrated reduced sucrose preference in the SPT and increased immobility time in the TST compared to the control group (Fig.1E–G). The FST results revealed that the immobility periods of the MPb and HPb groups were significantly longer than those of the control group (Fig. 1H). Additionally, the activity of mice in the LPb and MPb groups was diminished in the FST compared to that in the control group (Fig. 1I). These results demonstrate that Pb exposure induces depression-like behaviours in mice. To assess the long-term effects of Pb exposure induced depression-like behaviour in mice, we examined the immobility time of FST and sucrose preference for 12 and 24 w. The results demonstrated that Pb exposure induced a sustained reduction in sucrose preference rate in mice up to 24 w, whereas the immobility time in the FST peaked at 12 w (Supplementary Fig. 2A, B).

The ferroptosis indices were measured in the prefrontal cortex, and hippocampus of mice following Pb exposure. Increased $Fe^{2+}$ and MDA contents were observed in the prefrontal cortex, and hippocampus, whereas decreased GSH content was observed in the prefrontal cortex following Pb exposure, particularly in the MPb and HPb groups (Fig. 1J–L). The protein expression levels of GPX4 significantly decreased, whereas that of PTGS2 increased in the prefrontal cortex, and hippocampus of the MPb and HPb groups (Fig. 1M, N). Transmission electron microscopy indicated that Pb exposure decreased mitochondrial cristae, reduced mitochondrial volume, and vacuolisation of mitochondria in the prefrontal cortex, and hippocampus of mice (Fig. 1O, P). Correlation analysis demonstrated that ferroptosis indices in the prefrontal cortex, and hippocampus of mice following Pb exposure were associated with depression-like behaviours. The highest correlation coefficients between ferroptosis indices and depression-like behaviours were in the prefrontal cortex (Fig. 1Q, R). These findings suggest that Pb can induce ferroptosis in the mouse brain tissue, which is related to depression-like behaviours. In addition, the concentrations of $Pb^{2+}$ in the blood and prefrontal cortex of the LPb, MPb, and HPb groups were also measured. Thus, the concentrations of $Pb^{2+}$ in blood of LPb, MPb, and HPb groups were 69.23, 127.93, and 208.23 μg/L, respectively, whereas the prefrontal cortex and hippocampus $Pb^{2+}$ levels reached to 0.82 ± 0.24, 1.47 ± 0.23, and 2.01 ± 0.39 μg/g, respectively in prefrontal cortex, and 0.34 ± 0.14, 0.91 ± 0.21, and 1.41 ± 0.21 μg/g in hippocampus

(Supplementary Fig. 1). Fer-1, an inhibitor of ferroptosis, was used to further explore the role of ferroptosis in Pb exposure-induced depression-like behaviour in mice. Fer-1 treatment in the MPb group partially reversed the depression-like behaviour in mice. (Supplementary Fig. 2C). Collectively, these results suggest that ferroptosis is involved in Pb-induced depression-like behaviours.

### Astrocyte is identified as the most sensitive cell type to ferroptosis in mice brains following Pb exposure

To investigate the degree of ferroptosis sensitivity of different cell types in mouse brain tissue following Pb exposure, scRNA-seq and bioinformatic analyses were conducted to compute the ferroptosis score in different cell types following Pb exposure (Fig. 2A). A total of 78,448 individual cells were collected from the brain tissues of all groups. Of these, 42,341 cells were collected from the control group and 36,107 cells were collected from the Pb groups. The median UMI in the control and Pb groups were 3940 and 4243, respectively. By aggregation and quality control of the pre-processing, we obtained datasets of mouse brain tissues containing 32,770 and 32,232 cell profiles in the control and Pb groups, respectively (Supplementary Fig. 3A, B). Post integrated analysis, the obtained cells were arranged using uniform manifold approximation and projection (UMAP) in two dimensions for visualisation after standardization analysis (Fig. 2B). Unsupervised clustering revealed 18 distinct clusters across all samples. These clusters were identified as astrocytes, microglia, neurones, oligodendrocytes, epithelial cells, endothelial cells, neuronal stem cells, ependymal cells, neutrophils, and smooth muscle cells (SMC) based on the expression of known cell type markets (Fig. 2C) (Supplementary Fig. 3C, D). Additionally, the proportion of different cell types were computed. Overall, astrocytes, microglia, neurones, oligodendrocytes, epithelial cells, and other cells in the mice brain were 38.66, 40.89, 2.04, 7.61, 3.93, and 6.87%, respectively (Fig. 2D). The number of astrocytes and microglia in the Pb group increased, whereas the number of oligodendrocytes and epithelial cells decreased compared to that in the control group (Fig. 2E). Gene set score analysis revealed that Pb exposure increased the ferroptosis scores of astrocytes, microglia, and epithelial cells. Astrocytes demonstrated the highest ferroptosis scores among all cell types, which were 1.25- and 1.21-fold higher than those of microglia and epithelial cells (Fig. 2F). The Gene Set Enrichment Analysis (GSEA) results also demonstrated that the enrichment score of ferroptosis in astrocytes were much higher than microglia or epithelial cells (Fig. 2G). In addition, the number of astrocytes with high PTGS2 expression levels in the prefrontal cortex increased following Pb exposure (Fig. 2H). These results suggest that astrocytes are the most sensitive cells to ferroptosis in the mouse brain tissue following Pb exposure.

To investigate the temporal transcriptional changes in ferroptosis-related genes in astrocytes following Pb exposure, we initially subclustered astrocytes and identified marker genes for diverse subpopulations. Thus, seven astrocyte subpopulations were identified, among which, increased expression levels of *Crym, Hopx*, and *Id3* was observed in cluster 0; *Sparc, Itih3*, and *Slc6a11* was observed in cluster 1; *Nupr1, Igfbp2*, and *Pmp22* was observed in cluster 2; *Cadm2, Slc1a2*, and *Slc1a3* was observed in cluster 3; *Mt3, Mt1*, and *S100a1* was observed in cluster 4; *Ith3, Cd38, A2m*, and *Tfrc* was observed in cluster 5; and *Apoe, Vim, Gfap*, and *Acsl1* was observed in cluster 6 (Fig.3A, B). An increase in the number of cells was observed in clusters 4, 5, and 6, whereas a decrease was observed in clusters 1, 2, and 3, following Pb exposure (Fig.3C). The ferroptosis scores of astrocyte clusters 5 and 6 were significantly higher than those of the other subpopulations (Fig. 3D). The marker genes of clusters 5 and 6 were related to depression, neurodegenerative diseases, ferroptosis, and inflammation, whereas those of cluster 3 were related to the homeostatic functions of astrocytes. Based on the pseudo-time trajectory analysis and scores, the results demonstrated a chronological order of differentiation from cluster 3 to 5 (branch 1) and cluster 3 to 6 (branch 2) as the two branches of terminal states (Fig. 3E–G). Furthermore, the DEGs related to ferroptosis were identified, and the dynamic changes in ferroptosis-related DEGs were computed in branches 1 and 2 following Pb exposure. Thus, the ferroptosis-related DEGs presented

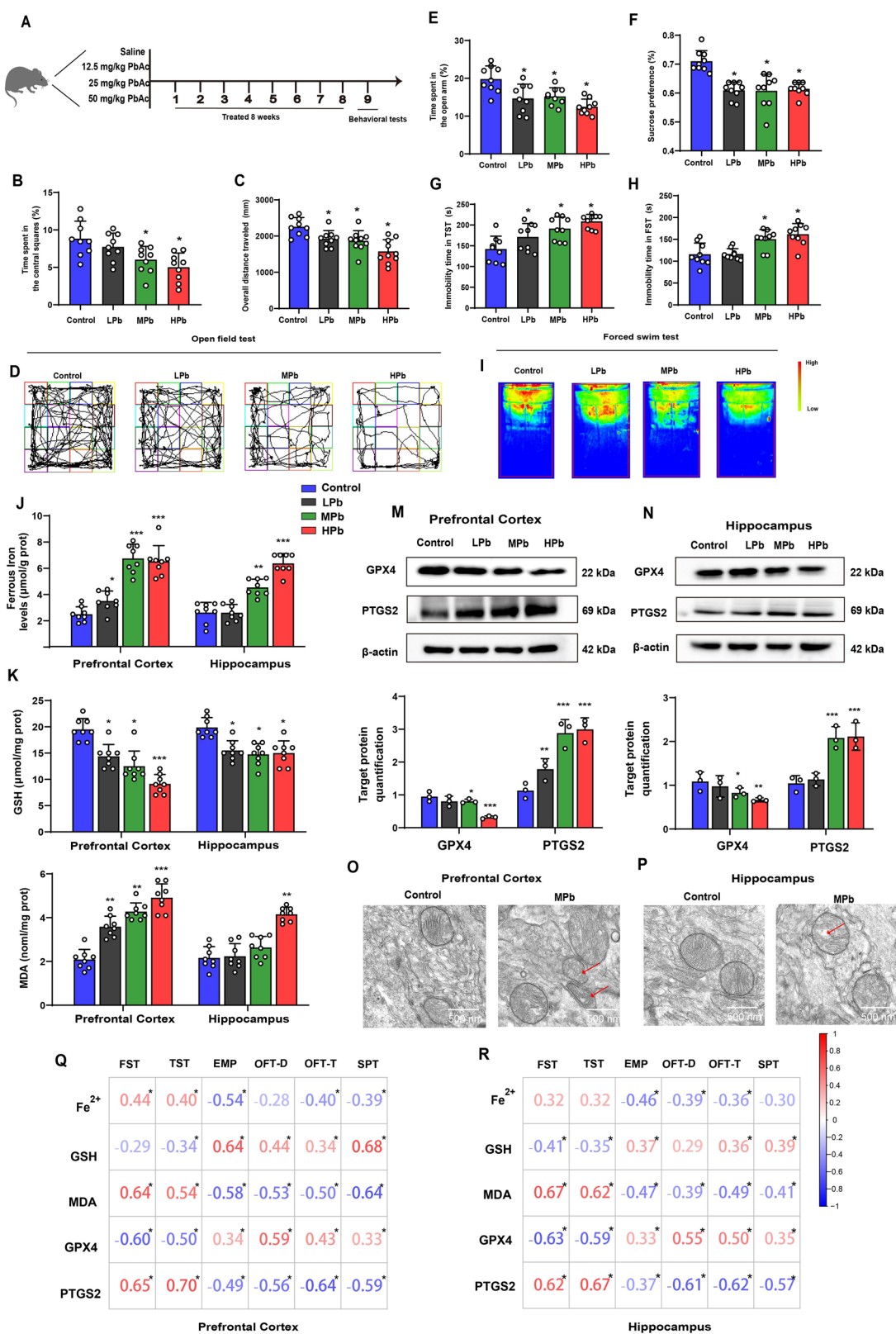

with temporal transcriptional changes following Pb exposure, which included *Vdac1, Vdac2, Fth1, Cisd1, Tfr, Tfrc, Iscu, Dmt1, Ftl1, Slc40a1, Cbs, Gstm1, Mgst1, Gsk3b, Slc3a2, Slc7a11, Gpx4, Hspa5, Sat1, Prdx1, Atf4, Nupr1, Atf3,* and *Hif-1α* (Supplementary Fig. 4) (Fig. 3H). These results suggest that ferroptosis-related genes are presented in astrocytes with changes in duration following Pb exposure.

## Perturbation of HIF-1α is an early effect in astrocytic ferroptosis following Pb exposure

We first isolated primary astrocytes from the mouse brain cortex and treated them with different doses of Pb and inhibitors of programmed cell death, including Z-VAD-FMK (an apoptosis inhibitor), Nec (a necroptosis inhibitor), Fer-1 (a ferroptosis inhibitor), DFO (a ferroptosis inhibitor), or

**Fig. 1 | Ferroptosis-related biological processes and protein are associated with anxiety- and depression-like behaviour in mice brain tissues following Pb exposure. A** Flow chart of animal treatment and behavioural test. SPF male C57BL/6 J mice are allocated into control, LPb, MPb, and HPb groups. After 8 weeks of Pb exposure, the depression-like behaviour are measured. **B** Time spent in the central squares and **C**. Overall distance travelled of control, LPb, MPb, and HPb groups in OFT ($n = 10$); **D** Moving trajectory of control, LPb, MPb, and HPb groups in OFT ($n = 10$); **E** Time spent in the open arm by the control, LPb, MPb, and HPb groups in EPM ($n = 10$); **F** Immobility duration of control, LPb, MPb, and HPb groups in TST ($n = 10$); **G** Sucrose preference of control, LPb, MPb, and HPb groups; **H** Immobility duration and (**I**) activity of control, LPb, and HPb groups in FST; The contents of (**J**). Fe$^{2+}$, **K** glutathione (GSH), **L** malondialdehyde (MDA) in the prefrontal cortex, hippocampus, hypothalamus, and striatum of different groups; Gel electrophoresis plot and semi-quantification of GPX4 and PTGS2 in (**M**). the prefrontal cortex, **N** hippocampus of mice ($n = 3$); **O**, **P** Ultrastructure of mitochondria in MPb group is imaged by transmission electron microscopy. The correlation analysis of the depression-like behaviour and ferroptosis-related biological process and proteins in the (**Q**). prefrontal cortex, and (**R**). hippocampus; $^*P < 0.05$ vs Control; $^{**}P < 0.01$ vs Control; $^{***}P < 0.001$ vs Control. Comparisons are made using ANOVA, followed by Tukey's multiple comparison test.

3-MA (an autophagy inhibitor) to explore whether ferroptosis could be the major cause of death in astrocyte (Supplementary Fig. 5A). The results demonstrated that PbAc reduced the astrocyte survival rate in a dose-dependent manner (Supplementary Fig. 5B, C). Meanwhile, DFO and Fer-1 significantly enhanced astrocyte viability following Pb exposure (Fig. 4A). These results suggested that ferroptosis is an important type of death in astrocytes following Pb exposure. Meanwhile, the levels of Fe$^{2+}$, MDA, LipROS, protein expression of PTSG2 increased, and the GSH content, GPX4 decreased. Additionally, demonstrated reduced or absent mitochondrial cristae and ruptured outer membranes were found in astrocytes by transmission electron microscopy (Fig. 4B–J). These results suggest that Pb exposure induces astrocytic ferroptosis in vitro. To further investigate the relative contributions of ferroptosis and apoptosis in Pb-induced astrocyte death, we examined the expression of apoptosis-related proteins. The results demonstrated that 5 μM Pb exposure did not alter the expression of apoptosis-related protein Caspase-3 in astrocytes. Notably, in 10 μM, 20 μM groups, the fold-changes in apoptotic proteins were substantially lower than PTGS2 protein expression,a protein related ferroptosis. These findings suggest that ferroptosis plays a more significant role than either apoptosis or necrosis in Pb-induced astrocyte death (Supplementary Fig. 5D).

To explore the early driver genes of astrocytic ferroptosis following Pb exposure, the perturbation analysis of sc-RNAseq were used to screen the DEGs of ferroptosis related genes. The results showed that *Hif-1α* had the highest AUC score, which suggested that *Hif-1α* are earliest perturbations gene compared with others DEGs of ferroptosis (Fig. 5A). The GSEA results also demonstrated that the HIF-1α signalling pathway was upregulated in the astrocytes following Pb exposure (Fig. 5B). Subsequently, HIF-1α expression was detected in vitro and in vivo. As expected, the *Hif-1α* mRNA expression levels was increased in the prefrontal cortex following Pb exposure. Moreover, the protein expression of nuclear and cytoplasmic HIF-1α was upregulated in astrocytes following Pb exposure. Compared with the control group, 5, 10, and 20 μM PbAc group exhibited 2.32-, 3.71-, and 4.09-fold increases in HIF-1α protein levels of the nucleus, and 1.41-, 2.28-, and 2.31-fold elevations in the cytoplasm, respectively (Fig. 5C, D). We further explore whether HIF-1α plays a critical role in astrocytic ferroptosis following Pb exposure. The HIF-1α knockdown astrocytes increased the GPX4 protein expression, and GSH levels, whereas suppressed the accumulation of Fe$^{2+}$, MDA, lipid peroxidation, levels, and PTGS2 protein expression in astrocytes following Pb exposure. Additionally, mitochondrial morphology was improved in the si-*Hif-1α* + Pb group compared to the Pb group (Supplementary Fig. 6) (Fig. 5E–M). Collectively, these findings suggest that *Hif-1α* may the early driver feature genes regulating the astrocyte ferroptosis following Pb exposure.

## Pb$^{2+}$ can inhibit HIF-1α degradation by directly binding to HIF-1α

HIF-1α is unstable and rapidly degraded by the hydroxylation of specific proline residues in normal oxygen state. To investigate the direct impact of Pb$^{2+}$ on HIF-1α and to thoroughly analyse the underlying molecular mechanism, we conducted molecular docking and molecular dynamics (MD) simulations. The molecular docking results showed that Pb$^{2+}$ has the strong affinity of HIF-1α in 138 residues of proline hydroxylation (PRO), 136 residues of threonine, 133 residues of phenylalanine and 141 residues of histidine (HIS), which the intramolecular distance is less than 5 Å (Fig. 6A). PRO138-141 is highly conserved in HIF-1α orthologues in human, mouse,

rat, and bovine, indicating that it is essential for the function of HIF-1α (Fig. 6B). We further investigated the specificity of the binding sites and compared with other metal ions, including Fe$^{3+}$, Zn$^{2+}$, and Mn$^{2+}$. The results showed that binding energies of the other three metal groups near the PRO138-HIS141 were weaker compared to Pb$^{2+}$, suggesting that Pb$^2$ and HIF-1α may exhibit specificity near this site (Supplementary Fig. 7A, B). The docked molecular conformation was preserved, and its molecular dynamics were simulated to gain insight into the dynamic process of Pb$^{2+}$-HIF-1α complex system. The mean interaction energy ($-746.27 \pm 145.90$) kcal/mol of 100 ns indicated that Pb$^{2+}$ tightly binds to HIF-1α (Fig. 6C). To explore the stability of Pb$^{2+}$-HIF-1α protein complex, Radius of gyration (Rg), root-mean-square-deviation (RMSD) and root mean square function (RMSF) was calculated. The Rg of the HIF-1α protein and the Pb$^{2+}$-HIF-1α complex system range from 2.27 to 2.77 nm and from 2.49 to 2.67 nm, respectively. Initially, the Rg slightly decrease and then show reduced fluctuations, gradually converging. At 100 ns, the Rg values are 2.37 nm and 2.54 nm, respectively. The RMSD of the system experience slight fluctuations during the first 20 ns of the simulation. However, as the simulation time increases, the amplitude of the RMSD fluctuations decreases, gradually stabilizing. Ultimately, the RMSD values are 0.94 nm and 0.38 nm, respectively. Subsequently, the RMSF of different PRO residue in HIF-1α was analysed. As a results, the RMSF of Pb$^{2+}$-HIF-1α complex system was reduced compared with HIF-1α protein system, which suggested that the PRO residues become more rigid (Fig. 6D–G). We measured the change in distance between three residues at the binding site, and the analysis at 0 and 100 ns confirmed Pb$^{2+}$-induced conformational changes in the HIF-1α protein, leading to stabilize HIF-1α proteins (Fig. 6H). Additionally, the Pb$^{2+}$-HIF-1α protein binding situation was also evaluated in vitro. The results showed that the colocalization area of Pb$^{2+}$-HIF-1α protein was increased (Fig. 6I). HIF-1α protein was further isolated from E. coli lysates overexpressing HIF-1α, and its purity was verified by SDS-PAGE (Supplementary Fig. 7C). The interaction between Pb$^{2+}$ and HIF-1α was investigated using ultraviolet (UV) spectroscopy, fluorescence spectroscopy (FS), and circular dichroism (CD) spectroscopy. The results showed that the intensity of the ligand-to-metal charge transfer band progressively intensified as the concentration of Pb$^{2+}$ increase, exhibiting a dominant peak at 205 nm and a shoulder peak at 225 nm (Fig. 6J). FS revealed that the fluorescence intensity of the Pb-HIF-1α complex increased with increasing concentrations of Pb$^{2+}$, indicating an interaction between Pb$^{2+}$ and HIF-1α (Fig. 6K). Additionally, upon the addition of Pb$^{2+}$, the negative peak intensity in the CD spectrum of HIF-1α increased, suggesting a rise in α-helical content. This structural alteration of HIF-1α likely arises from the binding of positively charged Pb$^{2+}$ to negatively charged residues in HIF-1α, which modifies its secondary structure via non-covalent interactions and enhances protein stability by facilitating α-helix formation (Fig. 6L). In summary, our results suggested that Pb$^{2+}$ could bind to residues of PRO in HIF-1α and inhibiting HIF-1α degradation.

## HIF-1α mainly regulate mitochondrial dysfunction promoting Pb-induced astrocyte ferroptosis by modulating VDAC1

To explore the mechanism of ferroptosis regulated by HIF-1α, the regulatory network of HIF-1α in astrocyte ferroptosis following Pb exposure was constructed. As a result, 65 potential gene were identified by KnockTF, hTFtarget, CHIP_Atlas, GTRD, CHEA database (Fig. 7A). DEGs of astrocytes ferroptosis were matched 65 potential gene to explore the

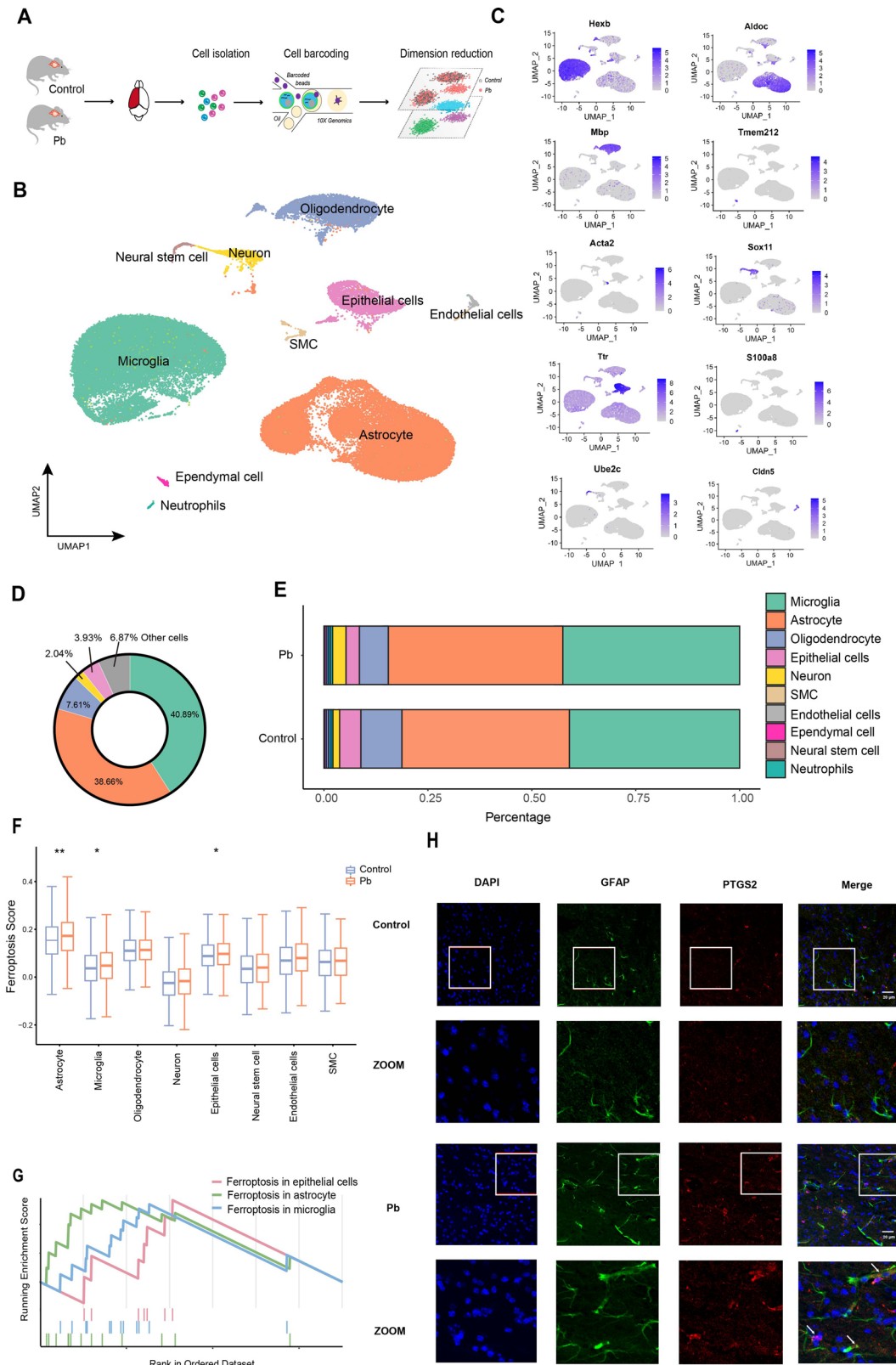

**Fig. 2 | scRNA-seq reveals that astrocyte is most sensitive cell type to ferroptosis following Pb exposure. A** Flow chart of scRNA-seq in mice brain tissue. Mice are allocated into Control and MPb groups. After 8 weeks of Pb exposure, the brain tissue of each group is extracted ($n = 3$). scRNA-seq is conducted on the single cells using 10×Genomics; **B** The UMAP plot demonstrating the annotation and colour codes for different cell types in Control and MPb groups ($n = 3$); **C** The representative marker genes of different cell types ($n = 3$). Dot size represents the proportion of cells expressing the gene, whereas the colour intensity represents the

average expression; **D** The proportion of all cell types in Control and MPb groups; **E** The proportion of all cell types in Control and MPb groups ($n = 3$); **F** The ferroptosis score of the cell types in different groups by AddModuleScore; **G** The enrichment score of ferroptosis in different cell types by GSEA.
**H** Immunofluorescence assays are conducted to detect the GFAP[+] and PTGS2[+] astrocytes in the Pb-exposed and control groups ($n = 3$). [*]$P < 0.05$ vs Control; [**]$P < 0.01$ vs Control. Comparisons between the two groups are made using Student's t-test.

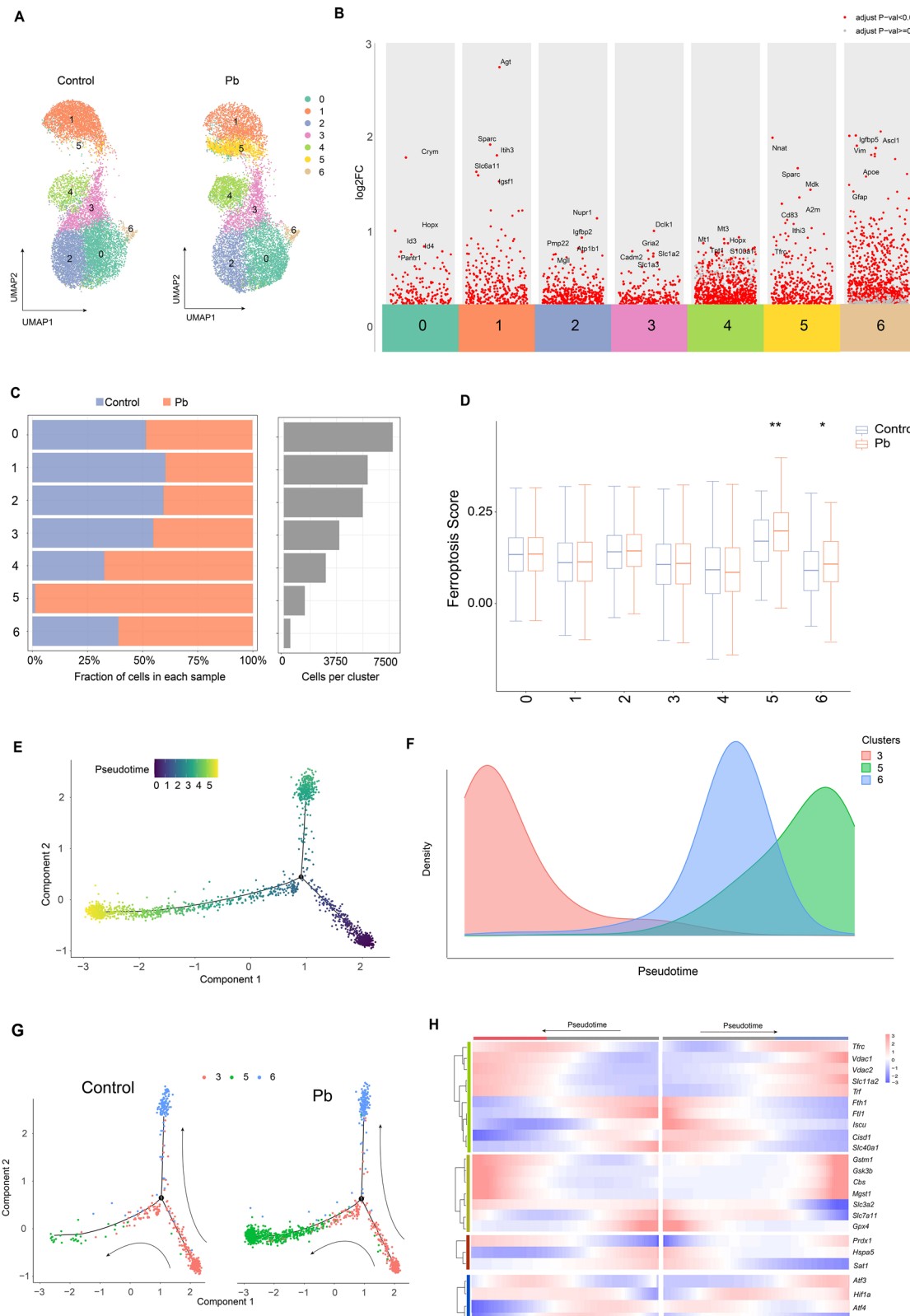

**Fig. 3 | Dynamic changes of ferroptosis related genes in astrocytes following Pb exposure. A** UMAP plot depicting the unsupervised astrocyte subpopulations in Control and MPb groups (n = 3); **B** Marker gene of astrocyte subpopulations following Pb exposure in astrocytes; **C** Proportion of astrocyte subpopulations in Control and MPb groups (n = 3); **D** The ferroptosis score of astrocyte subpopulations by AddModuleScore; **E, F** Pseudotime scores in differentiation state (cluster 3 to cluster 5 or 6) of astrocytes following Pb exposure from the lowest to the highest.

**G** Results of pseudotime trajectory analysis are split by exposure status and coloured based on cluster of astrocytes (red, green, and blue are represented cluster 3, 5, and 6), respectively; **H** Expression dynamics of ferroptosis-related DEGs in pseudotime trajectory in differentiation state of astrocytes (left represents cluster 3 to 5; right represents cluster 3 to 6). $^*P < 0.05$ vs Control; $^{**}P < 0.01$ vs Control. Comparisons between the two groups are made using Student's t-test.

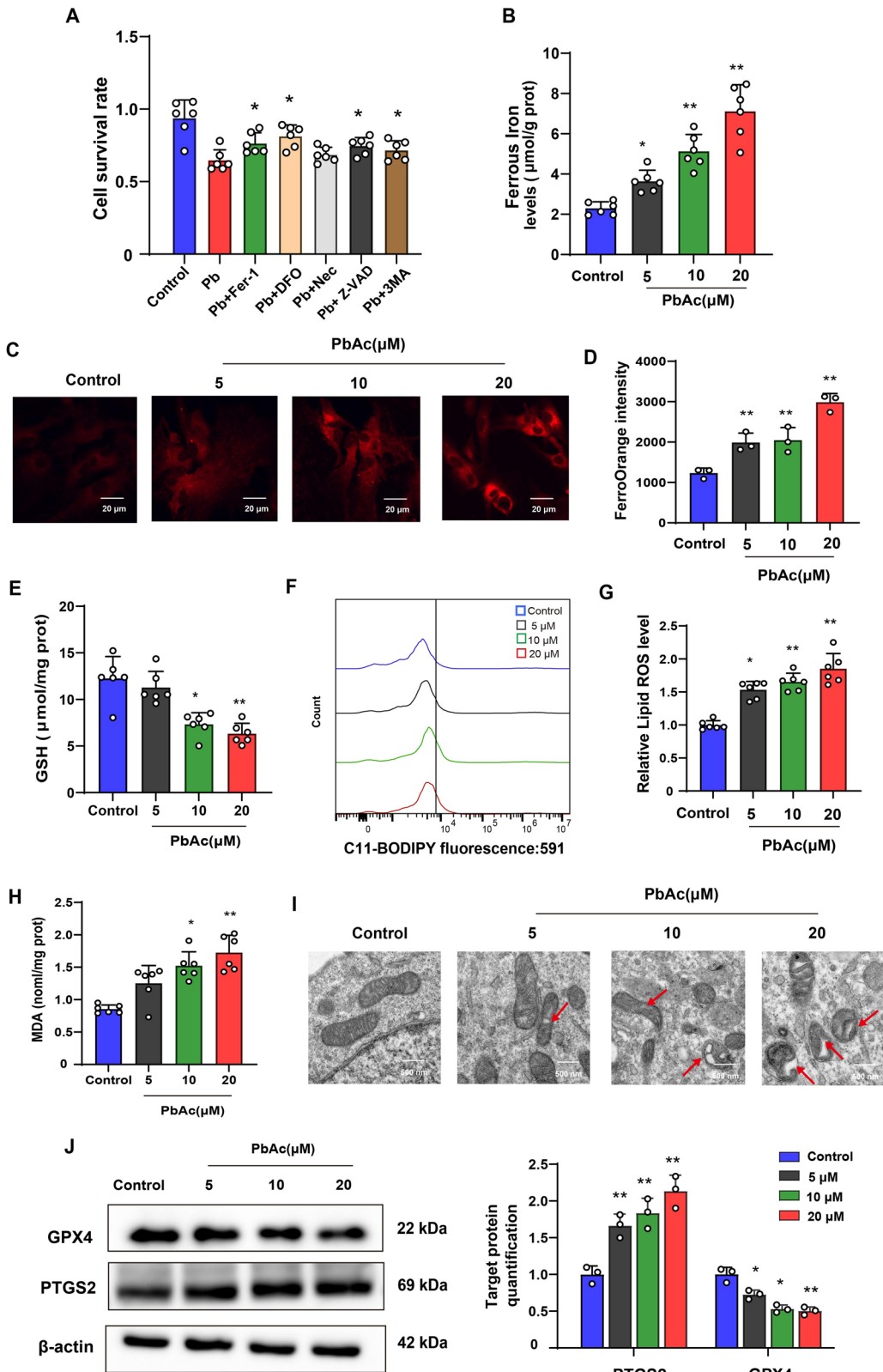

**Fig. 4 | Pb exposure-induced astrocyte ferroptosis in vitro. A** Cell viability of astrocytes in different Pb groups at 24 h ($n = 6$); **B**. $Fe^{2+}$ concentration in astrocytes following Pb exposure ($n = 6$); **C** $Fe^{2+}$ visualised by FerroOrange staining in astrocytes following Pb exposure; **D** Fluorescence intensity of FerroOrange in different groups analysed by ImageJ; **E** GSH content in astrocytes following Pb exposure ($n = 3$); **F** Fluorescence peak of LipROS in astrocytes following Pb exposure ($n = 6$); **G** Fluorescence intensity of C11-BODIPY in different groups ($n = 6$); **H** MDA content in astrocytes following Pb exposure ($n = 6$); **I** Ultrastructure of mitochondria in astrocytes of different groups captured using transmission electron microscopy ($n = 3$). Red arrowheads: shrunken mitochondria; **J** Gel electrophoresis plot and semi-quantification of GPX4 and PTGS2 in astrocytes following Pb exposure ($n = 3$); $^*P < 0.05$ vs Control; $^{**}P < 0.01$ vs Control. Comparisons are made with ANOVA, followed by Tukey's multiple comparison tests.

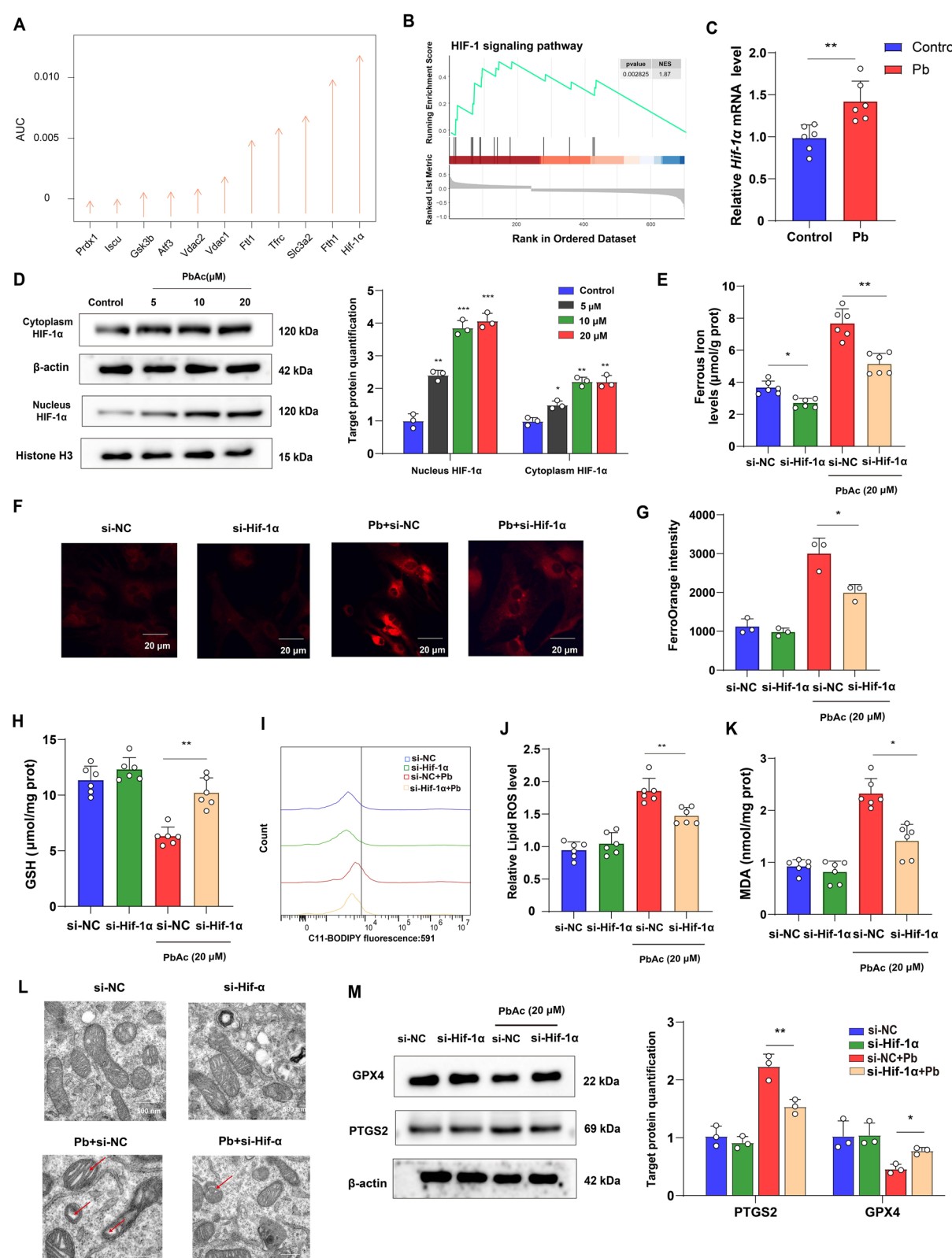

mechanism of HIF-1α regulated ferroptosis. The analysis revealed that potential two gene were related to astrocytic ferroptosis, namely VDAC1, and VDAC2 (Fig. 7B). Genotype–Tissue Expression (GTEx) database showed that the regulation of VDAC1 and VDAC2 by HIF-1α is highly specific in the brain (Fig. 7C). Meanwhile, the change in correlation coefficient is significant based on post-mortem human brains (Fig. 7D).

We further explore the regulator of HIF-1α following Pb exposure. Pb exposure increased VDAC1 and VDAC2 expression levels in astrocytes. Among these, the protein expression of VDAC1 demonstrated the highest change compared to that of VDAC2 (Fig. 7E, F). To explore whether HIF-1α regulated the increase of *Vdac1 expression* following Pb exposure, *Hif-1α* was knocked down in astrocytes, which reversed *Vdac1* upregulation

**Fig. 5 | *Hif-1α* knockdown rescued Pb-induced ferroptosis in astrocyte. A** The AUC score of ferroptosis DEGs following Pb exposure in astrocyte by Augur of sc-RNAseq; **B** Enrichment score of HIF-1α signalling pathway following Pb exposure; **C** mRNA expression of *Hif-1α* in mice brain tissues following Pb exposure; **D** Gel electrophoresis plot and semi-quantification of HIF-1α in cytoplasm and nucleus of astrocytes followed by different concentrations of PbAc (*n* = 3); **E** Content of Fe²⁺ in Pb-exposed astrocytes infected with si-NC and si-*Hif-1α* (*n* = 6); **F** Fe²⁺ visualised by FerroOrange staining in Pb-exposed astrocytes infected with si-NC and si-Hif-1α; **G** Fluorescence intensity of FerroOrange in Pb-exposed astrocytes infected with si-NC and Hif-1α analysed by ImageJ (*n* = 3); **H** GSH, **I** MDA content in Pb-exposed astrocytes infected with si-NC and si-*Hif-1α* (*n* = 6); **J** Fluorescence peak of LipROS by C11-BODIPY; **K** The fluorescence intensity of C11-BODIPY in Pb-exposed astrocytes infected with si-NC and si-*Hif-1α* (*n* = 6); **L** Ultrastructure of mitochondria in astrocytes of different groups was imaged by transmission electron microscopy. Red arrowheads: shrunken mitochondria; **M** Gel electrophoresis plot and semi-quantification of GPX4 and PTGS2 in Pb-exposed astrocytes infected with si-NC and si-*Hif-1α*, *$P < 0.05$ vs indicated group; **$P < 0.01$ vs indicated group; ***$P < 0.001$ vs indicated group. Comparisons are made with ANOVA, followed by Tukey's multiple comparison tests.

following Pb exposure (Supplementary Fig. 8) (Fig. 7G). *Vdac1*-related biological processes in astrocytes were assessed using Gene Ontology enrichment analysis and were predominantly enriched in mitochondrial dysfunction (Fig. 7H). In addition, *Vdac1* knockdown astrocytes were observed to suppress the mitochondrial increase in Fe²⁺ and mtSOX and increased the ATP content and MMP following Pb exposure (Fig. 8). These data suggest that VDAC1 is a critical protein related to mitochondrial dysfunction in the astrocyte regulating by HIF-1α.

Next, the *Vdac1* knockdown astrocytes increased GPX4 protein expression, and the content of GSH levels, whereas suppressed the accumulation of Fe²⁺, MDA, lipid peroxidation, levels, and PTGS2 protein expression in astrocytes following Pb exposure. In addition, the morphology of the mitochondria was also improved in the si-*Vdac1* + Pb group compared with that of Pb group (Fig. 9). Taken together, these finding suggest that mitochondrial dysfunction was involved in Pb-induced astrocyte ferroptosis regulating by HIF-1α/VDAC1.

## Discussion

Pb exposure is associated with depression-like behaviours. However, no well-established neural mechanisms have been elucidated. This study demonstrates that ferroptosis is involved in depression-like behaviours in mice following Pb exposure. Astrocytes are the most sensitive cells to ferroptosis in the mouse brain driving early by HIF-1α. Meanwhile, Pb²⁺ binds to HIF-1α, make HIF-1α stabilizion in terms of protein structure and inhibiting its degradation by occupying the proline hydroxylation sites. Further data demonstrated that HIF-1α mediated *Vdac1* regulating mitochondria dysfunction and promoting astrocyte ferroptosis following Pb exposure. These findings provide new insights into the mechanisms underlying Pb-induced depression-like behaviours.

Several studies have demonstrated that ferroptosis is involved in the pathogenesis of depression[21,22]. The number of astrocytes in the depression-related subpopulations increased following Pb exposure according to the scRNA-seq data. Additionally, ferroptosis-related genes were predominantly enriched in the astrocyte subpopulations associated with depression following Pb exposure. These findings suggest that astrocyte ferroptosis is an important cellular event in depression-like behaviours following Pb exposure. Pb has the ability to readily penetrate the blood-brain barrier and accumulate in brain tissue, where it triggers neuroinflammation. It activates the NF-κB transcription pathway, resulting in the upregulation of pro-inflammatory innate immune genes in astrocytes[23]. Additionally, Pb exposure can lead to the overactivation of microglia, causing an excessive release of inflammatory cytokines, which in turn contributes to neuronal inflammation, injury, and cell death[24,25]. Furthermore, ferroptosis is closely linked to the inflammatory responses in cells, ultimately leading to nerve damage[26]. Our scRNA-seq analysis of astrocytes revealed that inflammation-related marker genes, including IL-1β and IL-6, are expressed in ferroptosis-associated astrocyte subpopulations. This finding suggests a complex interplay between ferroptosis and inflammatory responses following Pb exposure. The data show that ROS levels increase, indicating that ROS production is a key regulatory factor in ferroptosis and inflammatory responses. Our findings were similar for microglia ferroptosis accompanying the high level of IL6 and IL-1β expression following Nitrogen-doped graphene quantum dots exposure[27]. Cell–cell communication regulates the activation of signaling pathways involved in cell function,

proliferation, and death. Microglia exposed to Pb secrete cytochemokines, resulting in the activation of caspase-3 and subsequent neuronal death[28]. Pb exposure induced microgliosis and astrogliosis in hippocampus of young mice potentially by triggering TLR4-MyD88-NF-κB signaling cascades suggesting that Pb could affect multiple cell types or signaling pathways induced the nerve injury[29]. Our results demonstrated that changes in cell counts were observed in the microglia and neurones following Pb exposure. This finding underscored that microglial and neuronal dysfunction could indirectly influence astrocytic ferroptosis through the release of pro-inflammatory cytokines or the activation of signaling pathways. The prefrontal cortex is one of the primary components of the limbic system, playing an irreplaceable role in learning, memory, planning, attention, and behavioral and emotional responses[30]. Our research findings showed that the expression of ferroptosis-related biological processes and marker proteins in the prefrontal cortex has the highest correlation with depression-like behaviors compared to the hippocampus. This suggests that the prefrontal cortex is more susceptible to ferroptosis than the hippocampus following Pb exposure in mice. Moreover, Pb exposure may primarily affect depression by inducing ferroptosis in the prefrontal cortex.

Cell viability was significantly inhibited by DFO, Fer-1, and Z-VAD-FMK, suggesting mixed cell death via ferroptosis and apoptosis. Pb exposure induces neuronal apoptosis by increasing oxidative stress and disrupting mitochondrial function[23]. Ferroptosis is intrinsically linked to oxidative stress, particularly the oxidative damage to lipids[31]. Lipid peroxides accumulation due to excessive ROS and iron is a hallmark of ferroptosis, which suggests that oxidative stress serves as a common upstream event that drives both apoptosis and ferroptosis through diverse mechanisms following Pb exposure. Under pathological scenarios ferroptosis often accompanies other cell death routines; however, inhibition of apoptosis or necroptosis is not sufficient to inhibit ferroptosis[32]. Previous studies have demonstrated that ferroptosis occurs before apoptosis in mouse models of Parkinson's disease[33]. This study demonstrated that Fer-1 can enhance the viability of astrocytes following Pb exposure than other inhibitors suggesting that ferroptosis plays a more critical role in the process of astrocyte death. We also observed that Fer-1 attenuated Pb-induced depression-like behaviour alterations. However, Fer-1 administration alone exerted no significant effects on depression-like behaviour. The observed lack of behaviour changes may be attributed to the administered dosage of Fer-1. In this study, 2 mg/kg Fer-1was used to animal experiment, which were based on the manufacturer's specifications and relevant literature (the recommended dosage range for Fer-1 in animal studies is 0.8–10 mg/kg). Wu et al study used 5 mg/kg Fer-1 with every fourth day for 28 days to explore the Nitrogen doped graphene quantum dots induced the learning and memory impairment. Sripetchwandee et al study used 2 mg/kg Fer-1 to explore the iron-overloaded induced cognitive disorder[34,35]. Notably, none of the doses within this range have been reported to induce behaviour changes in mice, which is consistent with our findings.

HIF-1α is a protein characterized by several key structural domains that play critical roles in its function. The N-terminal transactivation domain (TAD) is essential for recruiting co-activators and initiating the transcription of target genes. Meanwhile, the oxygen-dependent degradation (ODD) domain contains proline residues that undergo hydroxylation under normoxic conditions, which subsequently leads to the proteasomal degradation of HIF-1α[36]. Our results showed that the protein expression of HIF-1α and the

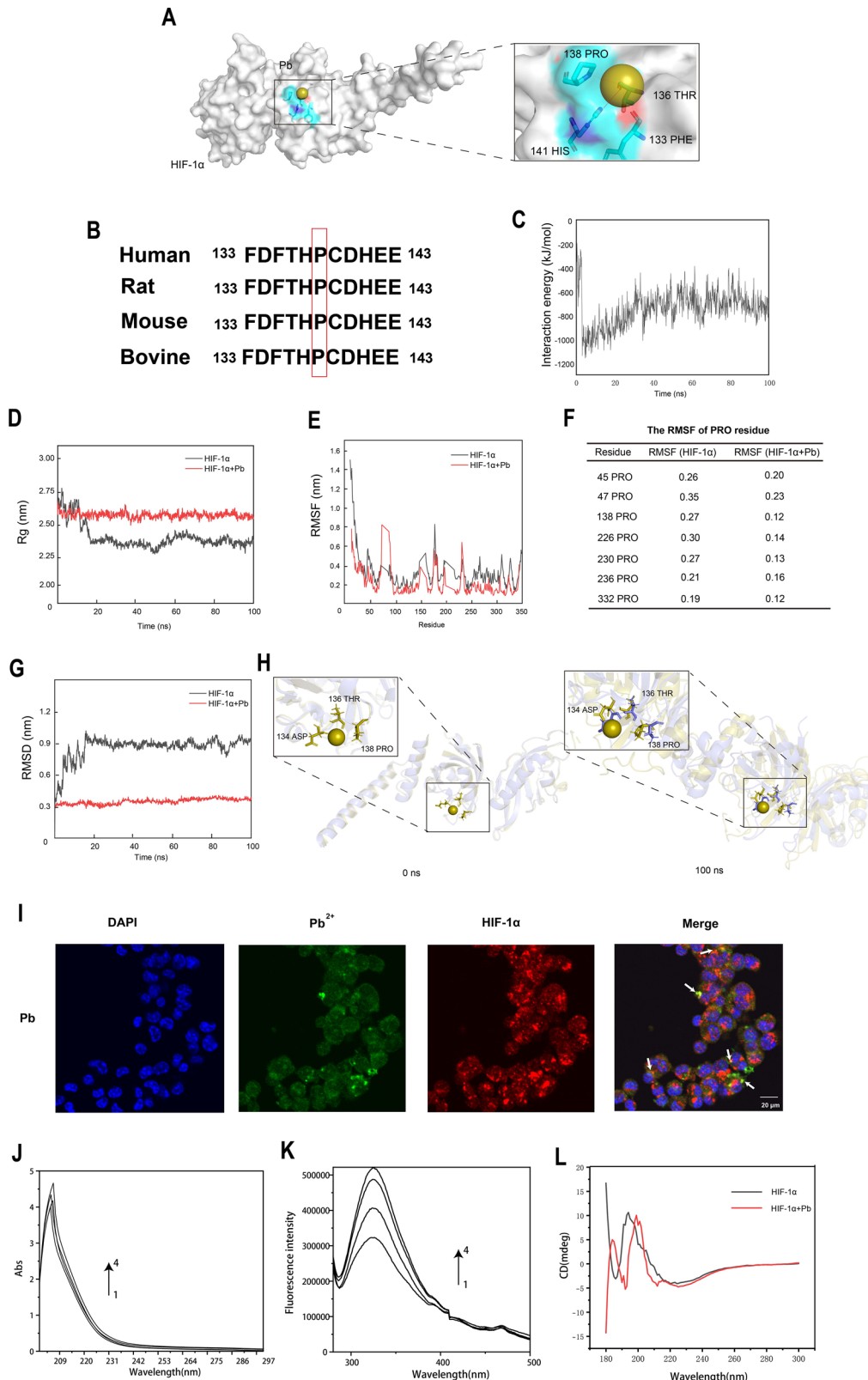

**Fig. 6 | Pb²⁺ can directly bind to HIF-1α inhibiting its degradation by Molecular docking and MD simulation. A** Molecular docking model illustrating the binding of Pb²⁺ to HIF-1α. Detailed interaction view (enlarged panel); **B** P138 of HIF-1α (marked in red) is conserved between different mammals; **C** Interaction energy between Pb²⁺ and HIF-1α; **D** The Rg of the HIF-1α protein and the Pb² + -HIF-1α complex system; **E**. The RMSF of the HIF-1α protein and the Pb² + -HIF-1α complex system; **F** The RMSF of the HIF-1α protein and the Pb² + -HIF-1α complex system in different PRO residue; **G** The RMSD of the HIF-1α protein and the Pb² + - HIF-1α complex system; **H** Conformational changes of the HIF-1α- Pb²⁺ complex in MD simulation; **I** Immunofluorescence assay was conducted to detect the Pb²⁺ and HIF-1α in astrocytes following Pb exposure; **J** UV-Vis absorption spectra, **K** Synchronous fluorescence spectra and (**L**) Circular dichroism spectrum of HIF-1α and Pb²⁺- HIF-1α systems (T = 298 K), C(HIF-1α) = 0.25 μM, C(Pb²⁺) (1 → 4): 0, 0.5, 1, 2 μM.

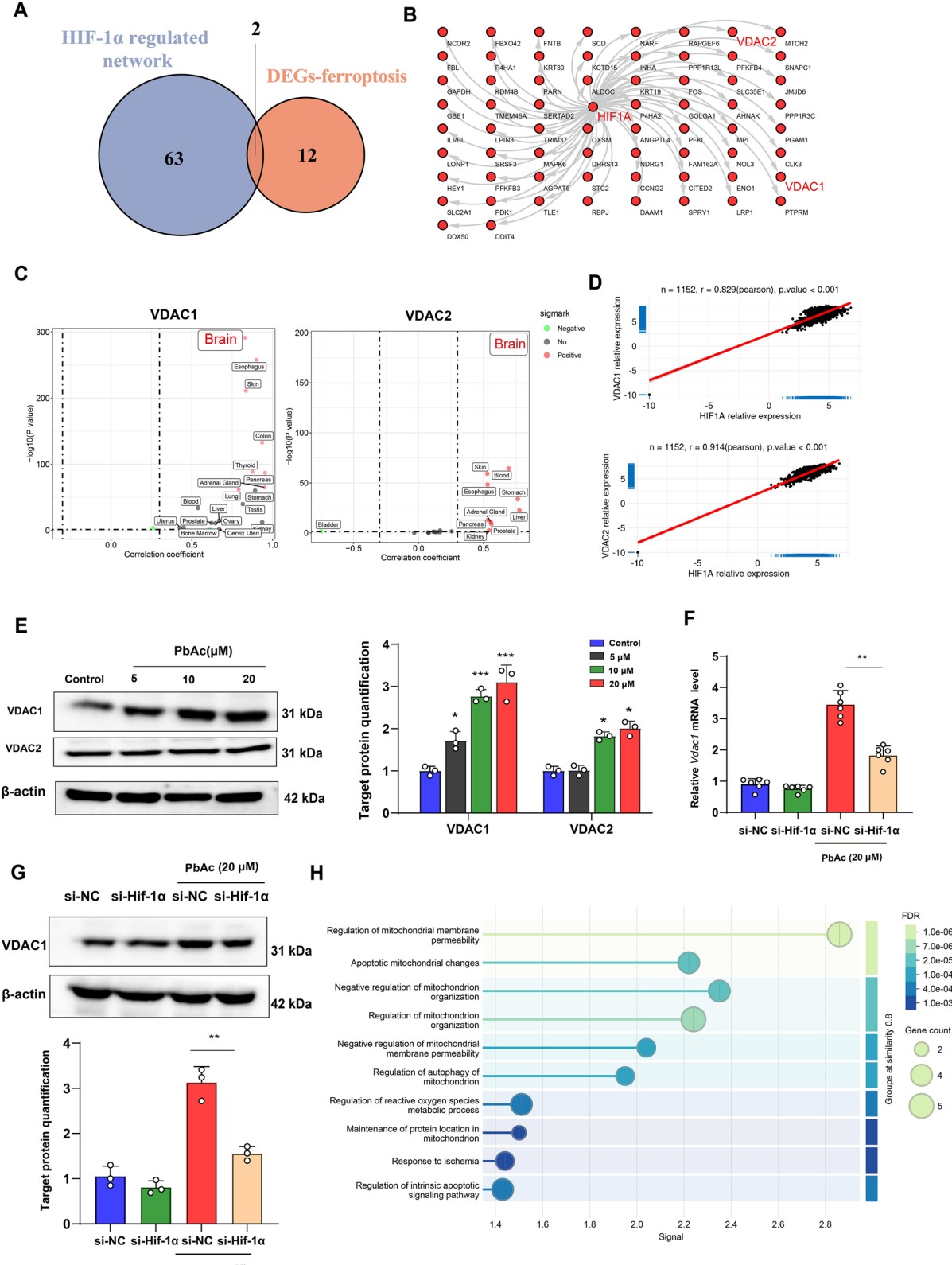

**Fig. 7 | HIF-1α mainly mediated *Vdac1* regulating in astrocyte ferroptosis following Pb exposure. A** Venn diagram of HIF-1α and DEGs-ferroptosis of astrocytes; **B** The networks of HIF-1α regulating in KnockTF, hTFtarget, CHIP_Atlas, GTRD, CHEA database; **C** VDAC1 and VDAC2 expression of different tissue based on GTEx database; **D** The correlation analysis of the *Hif-1α* and *VDAC1* and *VDAC2* based on GTEx; **E** Gel electrophoresis plot and semi-quantification of VDAC1, and VDAC2 in astrocytes following Pb exposure ($n = 3$); **F** Gene expression of *Vdac1* in Pb-exposed astrocytes infected with si-NC and si-*Hif-1α* ($n = 6$); **G** Gel electrophoresis plot and semi-quantification of VDAC1 in Pb-exposed astrocytes infected with si-NC and si-*Hif-1α* ($n = 3$); **H** GO enrichment analysis of VDAC1 proteins. $^*P < 0.05$ vs indicated group; $^{**}P < 0.01$ vs indicated group; $^{***}P < 0.001$ vs indicated group. Comparisons are made with ANOVA, followed by Tukey's multiple comparison tests.

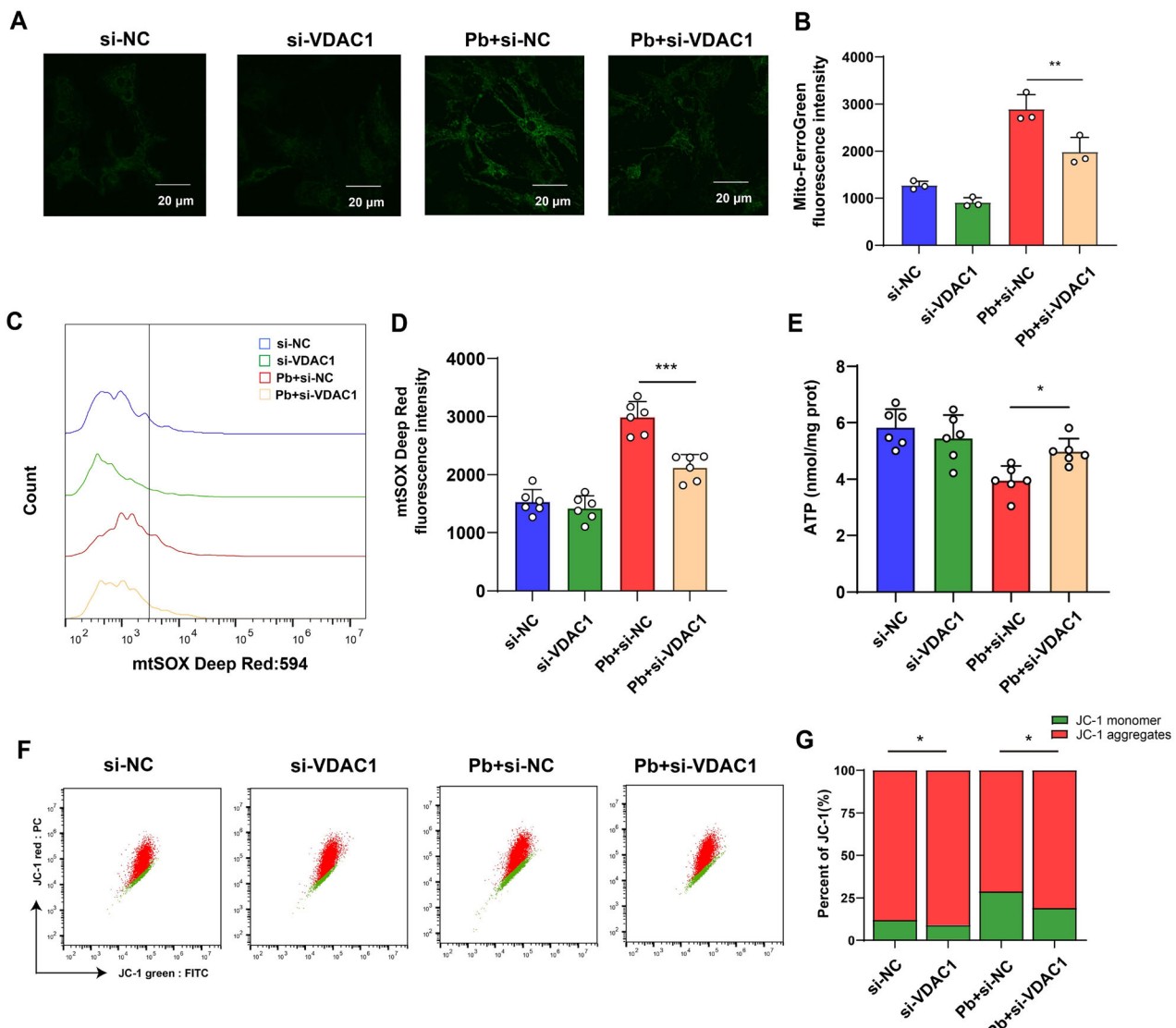

**Fig. 8 | *Vdac1* knockdown rescued Pb-induced mitochondrial dysfunction of astrocyte. A** Image of mitochondria Fe$^{2+}$ image by Mito-ferroGreen staining in Pb-exposed astrocytes infected with si-NC and si-*Vdac1*; **B** Fluorescence intensity of Mito-ferroGreen in Pb-exposed astrocytes infected with si-NC and si-*Vdac1* analysed by ImageJ ($n = 3$); **C** Fluorescence peak of mt-SOX in Pb-exposed astrocytes infected with si-NC and si-*Vdac1*; **D** Fluorescence intensity of mt-SOX in Pb-exposed astrocytes infected with si-NC and si-*Vdac1* ($n = 6$); **E** ATP content in Pb-

exposed astrocytes infected with si-NC and si-*Vdac1* ($n = 6$); **F** Dot plot of JC-1 monomer and aggregates in Pb-exposed astrocytes infected with si-NC and si-*Vdac1*; **G** Fluorescence intensity of JC-1 monomer and aggregates in Pb-exposed astrocytes infected with si-NC and si-*Vdac1* ($n = 6$); *$P < 0.05$ vs indicated group; **$P < 0.01$ vs indicated group; ***$P < 0.001$ vs indicated group. Comparisons are made using ANOVA, followed by Tukey's multiple comparison tests.

colocalization area of Pb$^{2+}$-HIF-1α protein were increased, suggesting Pb can directly bind to HIF-1α protein and inhibiting HIF-1α degradation.

Pb$^{2+}$, being divalent, have a strong tendency to coordinate with various ligands, including amino acid residues[37]. Proline, with its nitrogen-containing amine group and carbonyl oxygen in the backbone, provides suitable donor atoms for coordination[38]. Our data showed that the intramolecular distance between Pb$^{2+}$ and 138 PRO of HIF is less than 5 Å, while the mean interaction energy was increasing in Pb$^{2+}$ and HIF-1α complex system, suggesting the nitrogen atom in proline can form coordinate bonds with Pb$^{2+}$ facilitating the binding. HIF-1a have been identified as key regulator of cellular transcriptional programs in response to oxygen levels. HIF-1α can enter the nucleus and bind the multiple mRNA transcription and pathological process in oxidative stress and hypoxia, which contributed to cell ferroptosis[39,40]. Di-(2-ethylhexyl) phthalate exposure leads to ferroptosis through the HIF-1α/PTGS2 signalling pathway[41]. HIF-1α/SLC7A11 was involved in sorafenib attenuating liver fibrosis by triggering hepatic stellate cell ferroptosis[42]. Our data exhibited that inhibiting the *Hif-1α* expression

can reversed *Vdac1* upregulation following Pb exposure in astrocyte revealing that HIF-1α may regulate VDAC1 expression promoting ferroptosis of astrocyte.

Mitochondria promote ferroptosis through the regulation of iron metabolism, generation of reactive oxygen species, modulation of lipid metabolism, involvement in cellular signaling pathways, and maintenance of mitochondrial membrane integrity[43]. In states of mitochondrial dysfunction—such as oxidative stress, ischemia, or certain toxins—mitochondrial permeability transitions can occur. These transitions may involve the opening of the mitochondrial permeability transition pore, which allows for the passage of ions and small molecules, including excess iron and ROS, into the cytosolic space[44]. Our data demonstrated that mitochondrial Fe$^{2+}$ levels and mtSOX production increased in astrocytes following Pb exposure, whereas the MMPs and ATP was decreased, suggesting mitochondria dysfunction was involved in Pb-induced astrocyte ferroptosis. Our results showed that VDAC1 was the mainly targets protein regulated by HIF-1α following Pb exposure. VDAC1 is a pore-forming channel that regulates the

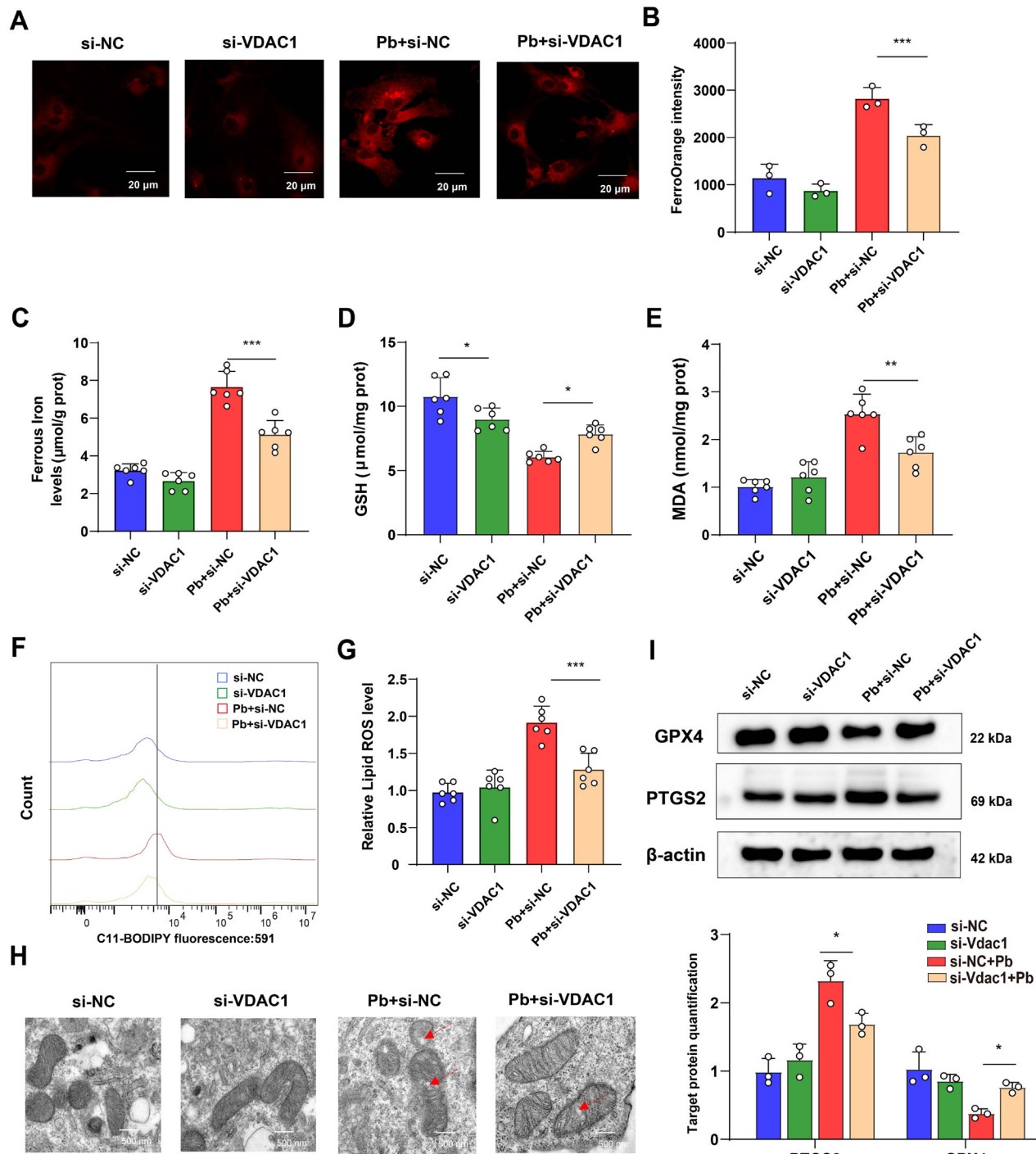

**Fig. 9 | *Vdac1* knockdown rescued Pb-induced ferroptosis in astrocyte.** A Fe²⁺ visualised by FerroOrange staining in Pb-exposed astrocytes infected with si-NC and si-*Vdac1*; **B** Fluorescence intensity of FerroOrange in Pb-exposed astrocytes infected with si-NC and *Vdac1* analysed by ImageJ (*n* = 3); **C** Content of Fe²⁺ in Pb-exposed astrocytes infected with si-NC and si-*Vdac1* (*n* = 6); **D** GSH, **E** MDA content in Pb-exposed astrocytes infected with si-NC and si-*Vdac1* (*n* = 6); **F** Fluorescence peak of LipROS by C11-BODIPY; **G** The fluorescence intensity of C11-BODIPY in Pb-

exposed astrocytes infected with si-NC and si-*Vdac1* (*n* = 6); **H** Ultrastructure of mitochondria in astrocytes of different groups was imaged by transmission electron microscopy. Red arrowheads: shrunken mitochondria; **I** Gel electrophoresis plot and semi-quantification of GPX4 and PTGS2 in Pb-exposed astrocytes infected with si-NC and si-*Vdac1*; [*]*P* < 0.05 vs indicated group; [**]*P* < 0.01 vs indicated group; [***]*P* < 0.001 vs indicated group. Comparisons are made with ANOVA, followed by Tukey's multiple comparison tests.

passage of many molecules and ions. Overexpression of VDAC1 can lead to the release of divalent iron and reactive oxygen species (ROS) from the mitochondria into the cytoplasm, promoting the development of ferroptosis[45]. Our data demonstrated that inhibiting of *Vdac1* suppressed the Pb-induced astrocyte ferroptosis, suggesting the alteration of

mitochondrial membrane permeability regulated by VDAC1 plays an important role in ferroptosis of astrocytes following Pb exposure.

This study has several limitations. One key limitation is the experimental design, which solely examined Pb exposure without including control groups for other heavy metals. Thus, it cannot be conclusively

established that Pb-induced depression-like behaviors are specifically mediated by ferroptosis.

In summary, we observed that ferroptosis is involved in Pb-induced depression-like behaviours in mice. Astrocytes are the most sensitive cell type to ferroptosis following Pb exposure in mouse brain tissues driving early by HIF-1α. Moreover, $Pb^{2+}$ binds to HIF-1α, stabilizing its protein structure and inhibiting its degradation by occupying the proline hydroxylation sites. Additionally, HIF-1α mediated *Vdac1* regulating mitochondria dysfunction and promoting astrocyte ferroptosis following Pb exposure. Our findings provide insights and perspectives for the investigation of Pb-induced depression-like behaviours.

## Method

### Animals and treatment
A total of 80 SPF male C57BL/6 J mice were obtained from Beijing Huafukang Biotechnology Co., Ltd. The mice were housed in a room with a completely controlled environment at constant temperature and humidity and a 12:12 h light/dark cycle, with free access to food and water. After adaptive feeding for a week, the mice were randomly allocated into control, Low Pb (LPb), Medium Pb (MPb), High Pb (HPb), ferroptosis inhibitor (Fer-1), and intervention (Pb+Fer-1) groups based on body weight. Mice in LPb, MPb, and HPb groups were administered 12.5, 25, and 50 mg/kg PbAc, respectively, by gavage for 8 weeks. Mice in Pb+Fer-1 group were administered 25 mg/kg PbAc by gavage and the Fer-1 group was intraperitoneally injected with 2 mg/kg Fer-1. The mice were treated with PbAc daily and Fer-1 every two days. Additionally, the control group was gavaged with an equal volume of saline solution. The Pb and Fer-1 dose was referenced to the previous studies[34,46,47]. All experiments were conducted in accordance with the National Institutes of Health Guide for the Care and Use of Laboratory Animals and approved by the North China University of Science and Technology Animal Ethics Committee (No. 2021-SY-201).

### Primary astrocyte cultures and treatment
Primary astrocytes were isolated from the cerebral cortex of the C57BL/6 J mice within 24 h of birth. C57BL/6 J mice were sacrificed and the cerebral cortices were collected. The meninges in the cerebral cortex were carefully stripped and immersed in the Hanks' buffered salt solution (0.5 mL). Five cerebral cortices were pooled and dissected into approximately 1 mm cubes. The blocks of cerebral cortices were digested using 0.25% trypsin-EDTA at 37 °C for 5 min followed by centrifugation at $1000 \times g$ for 5 min. The cells were enumerated, and the culture medium containing 90% Dulbecco's modified Eagle's medium, 10% foetal bovine serum, penicillin (100 U/mL), and streptomycin (100 μg/mL) was added. The suspension was then transferred into a polylysine-covered T25 cell culture flask with a volume of 25 cm². When the primary culture of cells was nearly confluent, the flasks were agitated at $250 \times g$ for 6 h to remove microglia and oligodendrocytes. Approximately 95% of these cells were positive for glial fibrillary acidic protein (GFAP), a marker of astrocytes cell types.

For each experiment using primary astrocytes, the plating density was $1 \times 10^4$ cells per well for 96-well plates and $5 \times 10^5$ cells per well for 6-well plates. The concentrations of PbAc used in the experiment were 5, 10, 20, 40, and 80 μM for 24 h, respectively. For the experiment which used primary astrocyte pre-treatment, 5 μM Fer-1, 10 μM Deferoxamine (DFO), 5 mM 3-methyladenine (3-MA), 1 μM necrosulfonamide (Nec), or 10 μM Z-VAD-FMK were used for 6 h prior to 20 or 80 μM PbAc treatment.

### Analysis of $Pb^{2+}$ in blood and brain tissues
Whole blood and brain tissue samples were digested with 2 mL ultrapure 65% $HNO_3$ using microwave digestion technique (CEM, USA). The digestion procedure was 100 °C constant heating for 2 h; 120 °C constant heating for 2 h; 140 °C constant heating for 3 h; 180 °C constant heating for 1 h Subsequently, the samples were placed on 110–150 °C graphite heating plates to remove the acid. After the samples were evaporated to approximately 0.1 mL, they were cooled to 25 °C. All the samples were diluted to 5 mL using distilled water. The concentration of $Pb^{2+}$ in the samples was quantified by inductively coupled plasma mass spectrometry (ICP-MS) (Agilent 8900, Agilent, USA), according to a previously described protocol. The limit of detection for Pb was 0.1 μg/L or 0.1 ng/g wet tissue.

### Behavioral tests
**Sucrose preference test**. The mice were individually housed and adapted to drinking 1% sucrose water for 2 days at the start of the experiment. During the first 24 h, two bottles containing 1% sucrose water were placed in each cage. In the following 24 h, one bottle was filled with 1% sucrose water and the other bottle with pure water. Bottle positions were switched for 12 h to avoid side biases. Subsequently, the mice were deprived of food and water for 12 h, and then they were given access to 1% sucrose solution and water for 12 h. The total liquid and sucrose water consumption of each mouse was measured after 12 h. Sucrose water preference was calculated as [(sucrose water consumption/total liquid consumption) × 100%].

**Forced swim test**. Each mouse was placed in a clear 3000 mL container filled with approximately 2000 mL of water at room temperature (22 °C) for 15 min. After 24 h, the mice were placed in the containers for 6 min. The duration of immobility, which includes staying afloat and attempts to stay afloat, was measured using SMART V3.0 (RWD Tech, Shen Zhen, China) during the last 5 min of the swimming time.

**Tail suspension test**. Each mouse was suspended by its tail with adhesive tape for 6 min and a video was recorded. The duration of immobility, when they hung passively and motionlessly for at least 2 s, was measured using SMART V3.0, during the last 5 min of the suspension time.

**Open field test**. The mice were placed in a square box (43 × 43 × 30 cm) in a brightly lit room for the open field test. The square was divided into 16 smaller squares of equal size. The mice were then placed in the square box. The movement around the square was video recorded for 5 min. After each trial, the test chambers were cleaned with dry tissue paper followed by 75% alcohol and cleaned with the tissue again. Distance travelled, time spent in the centre, and time spent in the total area were automatically measured for each mouse using SMART V3.0.

**Elevated plus maze test**. The maze was divided into four alternating quadrants, two of which had 14 cm high walls (closed arms) and two of which had no walls (open arms). The mice were placed in a closed arm and movements around the maze were recorded for 5 min. After each trial, the test chambers were cleaned with dry tissue paper followed by 75% alcohol and cleaned with the tissue again. The time spent in the closed and open arms was measured using SMART V3.0.

**Cell viability assay**. Cell viability was measured using a CCK-8 kit. Astrocytes were seeded in 96-well plates at a density of $1 \times 10^4$ cells per well. When the astrocyte cultures reached 80% density, they were treated with diverse concentrations of PbAc for 24 h, DFO, Fer-1, 3-MA, Nec, and Z-VAD-FMK for 6 h before the cells were incubated with CCK-8 solution. The absorbance (A) at 450 nm was quantified using an automated reader (Synergy HTK; BioTek, Winooski, VT, USA).

**$Fe^{2+}$, MDA, and glutathione quantification**. The mice brain tissue (prefrontal cortex, hippocampus) and astrocytes in different groups were homogenised by ultrasonication on ice for 3 min. This was followed by centrifugation at $12,000 \times g$ at 4 °C for 10 min. The supernatant was prepared according to the manufacturer's instructions to detect $Fe^{2+}$, GSH, and MDA contents (I291, A006-1-1, A003-1-2; Dojindo, Japan; Nanjing Jiancheng, China) by measuring the absorbance at 405, 532, and 593 nm, respectively. $Fe^{2+}$, GSH, and MDA contents were computed using a standard curve or standardised formula.

**Fe$^{2+}$ staining.** FerroOrange (F374, Dojindo, Japan) and Mito-FerroGreen (M489, Dojindo, Japan) were used to stain intracellular and mitochondrial Fe$^{2+}$ in astrocytes according to the manufacturer's instructions. Additionally, organelle-specific trackers and FerroOrange were used to explore Fe$^{2+}$ distribution in different organelles. The differently treated astrocytes were incubated with 1 mM FerroOrange, 5 μM Mito-FerroGreen working solution, or a combination of working solutions of different organelle-specific trackers and FerroOrange for 30 min at 37 °C. The cells were then observed under a confocal microscope (FV3000; Olympus, Tokyo, Japan) at 488 and 561 nm, respectively. The ImageJ software was used to measure the mean fluorescence intensity per unit area.

**Lipid peroxidation assay.** Intracellular and mitochondrial lipid peroxidation in astrocytes was measured using C11-BODIPY (D3861; Thermo Fisher Scientific, USA) according to the manufacturer's instructions. The differently treated astrocytes were incubated with C11-BODIPY for 30 min at 37 °C. Post staining, the cells were washed three times and resuspended in PBS. Cells were assayed using a flow cytometer at 594 nm (CytExpert1.2; Beckman, USA).

### Single-cell RNA sequencing

**Single-cell isolation and sequencing.** Three mouse brains from each of the control and Pb groups were collected for scRNA-seq analysis. scRNA-seq detection was conducted based on the workflow developed by SHBIO Biotechnology Limited Company (Shanghai, China). Briefly, the mouse brain tissue was digested to a single-cell suspension. Cell viability was evaluated by trypan blue staining and enumeration using a cytometer (Countess II, Thermo Fisher Scientific). Approximately 10,000 cells were captured in each sample with a viability of >80%. The libraries were prepared according to the manufacturer's protocol using Chromium Next GEM Single Cell 3′ GEM, Library & Gel Bead Kit v3.1 (10× Genomics). The libraries were then sequenced using a NovaSeq6000 (Illumina).

**scRNA-seq data processing.** Data were trimmed using the CellRanger software. FASTQs generated by Illumina sequencing were imported into CellRanger. The gene-barcode matrices for each individual sample were obtained by enumerating the unique molecular identifiers (UMIs) and filtering out any non-cell-associated barcodes that were output to the gene-barcode matrix. The gene-barcode matrix was analysed by the Seurat package of R (version 4.2.3) to conduct quality control and downstream analysis of scRNA-seq data. For quality control, cells with <500 or >6500 detected genes, >20% mitochondrial genes, and >20% ribosomal genes were excluded from further analysis. Genes detected in fewer than three cells were excluded. A total of 2000 highly variable genes were identified using the FindVariableFeatures function for downstream principal component analysis (PCA). To avoid confounding batch effects within and between the Pb and control groups, the data were adjusted for batch effects using the harmony package. PCA was used for dimensionality reduction, followed by clustering based on the K-nearest neighbour graph with optimal resolution. UMAP was used to visualise the results. The FindAllMarkers function (min.pct=0.2, Log |FC | ≥ 0.2. Threshold=0.2) was used to screen cell type markers in different clusters. The cell types for each cluster were manually annotated using previously published literature[48–50]. After verifying the cell type, the actual number and proportion of cells in the different groups were calculated. For the sub-astrocyte analysis, the cell types which were named "astrocyte" were be selected. The FindAllMarkers function was applied to screening the sub-astrocyte markers in different clusters.

### Bioinformatics analysis of scRNA-seq

**Differential expression gene screening and enrichment analysis.** Log |fold-change| (Log|FC | ) ≥ 0.2 and $P < 0.05$ were set as the thresholds for screening differentially expressed genes (DEGs) of different cell types between the control and Pb groups. Gene Ontology (GO) was conducted using the R package ClusterProfiler[51]. The cut-offs for differential biological process identification were set at $P < 0.05$, and |NES | > 1.

**The gene set scores analysis of ferroptosis.** FerrDb (http://zhounan.org/ferrdb/current/) and previously published results on ferroptosis genes were used to construct a gene set for ferroptosis. The AddModuleScore function of the Seurat package and GSEA were used to identify gene set scores of ferroptosis in the different cell types following Pb exposure.

**Prediction of potential gene regulating by *Hif-1α*.** The network of potential gene regulating by *Hif-1α* was constructed using the Shiny TF-Target Finder (https://jingle.shinyapps.io/TF_Target_Finder/)[52]. The final interaction network was obtained by taking the intersection with KnockTF, ENCODE, JASPAR, CHEA, GTRD, CHIP_Atlas, and hTFtarget. In addition, the Genotype-Tissue Expression was used to calculate the different tissue Correlation of HIF-1α/VDACs in different tissues including data from 1,152 human autopsy samples.

**Analysis of early Perturbed genes in ferroptosis following Pb exposure.** Prioritization of ferroptosis gene responsive to Pb exposure in single-cell data was calculated by Augur package[53]. AUC scores were evaluated the Perturbed sequence.

**Pseudotime trajectory analysis.** Pseudotime trajectory analysis of astrocytes was conducted using the Monocle 2 R package (version 2.26.0)[54]. Gene barcode matrices of different sub-astrocytes (500 cells in each cluster) were used as inputs. Astrocyte marker genes in each cluster were used to construct developmental trajectories. The final results were visualised by cell trajectory plots coloured according to cell state, Seurat clusters, and pseudotime scores using the plot_cell_trajectory function.

**Molecular docking.** Molecular docking was performed to investigate the binding interactions between Pb$^{2+}$, Fe$^{3+}$, Zn$^{2+}$, Mn$^{2+}$ and the HIF-1α protein. The sequence of HIF-1α protein was downloaded from uniport (https://www.uniprot.org/uniprot/), and the PDB file was processed by removing water molecules and cations before the subsequent docking step. AutoDock was performed the docking simulations. The docking environment was set up by defining the grid box around the binding site of HIF-1α, ensuring that it encompassed the regions where the Pb$^{2+}$, Fe$^{3+}$, Zn$^{2+}$, Mn$^{2+}$ were expected to interact. The docking simulations were conducted by placing Pb$^{2+}$, Fe$^{3+}$, Zn$^{2+}$, Mn$^{2+}$ in the defined grid box, which predicted the binding conformations and affinities of the Pb$^{2+}$, Fe$^{3+}$, Zn$^{2+}$, Mn$^{2+}$ with HIF-1α.

**MD simulations.** MD simulations were conducted to dissect the trajectories and interaction energies within the Pb$^{2+}$-protein complex. The optimal structures obtained from the molecular docking served as the starting conformations for MD simulations. After preprocessing the complex for simulation, Gromacs2022 was chosen as the MD simulation software, and Amber99sb-ildn was used as the protein force field. The TIP3P model was employed to add water to the system creating a water box of size $10 \times 10 \times 10$ nm$^3$, and ions were added to automatically balance the system. Particle-mesh Ewald was used to handle electrostatic interactions, and the steepest descent method was employed for energy minimization with a maximum of 50,000 steps. The cutoff distance for Coulombic interactions and the van der Waals radius cutoff distance was both set to 1 nm. Finally, the system was equilibrated using the canonical ensemble and the isothermal-isobaric ensemble, followed by 100 ns of MD simulation at constant temperature and pressure. The cutoff value for non-bonded interactions was set to 10 Å. The Langevin thermostat was used to control the simulation temperature at 300 K, and the Berendsen method was employed to control the pressure at 1 bar.

**Transmission electron microscopy**. The mice brain tissues and astrocytes in different groups were prefixed in 2.5% glutaraldehyde phosphate (0.1 M, pH 7.4) overnight at 4 °C, then postfixed in 2% buffered osmium tetroxide and embedded in Epon812 (Merck, Darmstadt, Germany) followed by dehydration. Ultrathin sections (60 nm thick) were cut and stained with uranyl acetate and lead citrate. Transmission electron microscopy (TEM) images were captured using a BioTEM system (JEM 1200EX, Tokyo, Japan).

**Cellular and tissue immunofluorescence**. Immunofluorescence staining was performed on cultured cells and mouse brain tissue sections from the different groups. Cells or sections were first fixed for 20 min using PBS containing 4% paraformaldehyde. Post three washes, the fixed cells or tissue sections were permeabilised with 0.4% Triton×100, for 30 min. Subsequently, they were blocked using a 3% blocking solution for 1.5 h. The cells were incubated with antibodies against HIF-1α (1:200) (Proteintech, Wuhan, China) at 4 °C overnight. The tissue sections were incubated with antibodies against PTGS2 (1:100) and GFAP (1:200) for two nights, followed by the appropriate Alexa Fluor 488- or 594-conjugated secondary antibodies at 37 °C for 1 h. Nuclei were stained using DAPI. Images were acquired using an Olympus FV3000 confocal microscope.

**ATP measurement**. ATP content of the samples was measured using the ATP assay kit according to the manufacturer's instructions (S0026; Beyotime, China). Differently treated astrocytes were lysed using ATP assay buffer, followed by centrifugation at $12,000 \times g$ at 4 °C for 5 min. The supernatant was used to determine the total ATP protein concentration. Assay buffer was added to each supernatant. Luminescence was recorded using an automated reader.

**Mitochondrial SOX assay**. The MitoSOX levels were measured mitochondrial superoxide detection kit according to the manufacturer's instructions (MT14; Dojindo, Japan). Differently treated astrocytes were collected and incubated with 10 μM MitoSOX probe for 30 min at 37 °C. Post staining, the cells were washed three times and resuspended in PBS. Cells were assessed using a flow cytometer at a wavelength of 594 nm.

**Mitochondrial membrane potential assay**. Mitochondrial membrane potential ($\Delta\Psi$m) in astrocytes was assessed using the JC-1 assay kit (C2003S; Beyotime, China) according to the manufacturer's protocol. Differently treated astrocytes were collected and added to the JC-1 working solution for 20 min. Post staining, the cells were washed using the JC-1 assay buffer three times and resuspended in JC-1 assay buffer. The cells were then assayed using a flow cytometer. The decrease in $\Delta\Psi$m was assessed by the transition from JC-1 aggregates (red fluorescence) to JC-1 monomers (green fluorescence).

**Real-time PCR**. Total RNA was extracted using TRIZOL. RNA was converted into complementary DNA (cDNA) using a cDNA Reverse Transcription Kit (Roche, Switzerland). β-actin was chosen as the reference gene (housekeeping). Amplification was conducted over 40 cycles. Primer sequences are provided in Supplementary Table 1.

**Western blotting**. Total protein was extracted using a whole-cell lysis assay kit (P0013B; Beyotime, China). Cytoplasmic and mitochondrial proteins were extracted using the Cytoplasmic and Mitochondrial Protein Extraction kit (C3601, Beyotime, China). Nuclear and cytoplasmic proteins were extracted using nuclear and cytoplasmic protein extraction kits (P0027; Beyotime, China). The entire process was conducted in strict accordance with the manufacturer's instructions. The protein concentration was determined using a BCA protein assay kit. The proteins were separated using 10% SDS-PAGE gel electrophoresis before they were transferred to a polyvinylidene fluoride membrane (Immobilon; Millipore, Burlington, MA, USA). The membranes were blocked with 5% skimmed milk in TBST and then sequentially reacted with monoclonal antibodies against GPX4 (1:1000), β-actin (1:3000), PTGS2 (1:500), HIF-1α (1:4000), HISTONE H3 (1:6000), VDAC2 (1:1000), Caspase 3 (1:1000) and VDAC1 (1:2000) (Proteintech, China), overnight at 4 °C. After washing with TBST, the blots were incubated with secondary HRP-linked anti-rabbit and anti-mouse IgG antibodies (Proteintech, China) at 37 °C for 2 h. The membranes were visualised using an enhanced chemiluminescence plus kit (P0018M; Beyotime, China). ImageJ software was used to measure the grey values of the target and internal reference protein bands. The ratio of the target protein band to the internal reference protein band was used as the final expression value.

**RNA interference**. *Hif-1α*, and *Vdac1* targeting siRNAs (si-*Hif-1α*, si-*Vadc1*) or nontargeting control siRNAs (si-NC) were synthesised by RiboBio (Guangzhou, China). Astrocytes were seeded in 6-well plates and transiently transfected with 20 μM PbAc, si-*Hif-1α*, si-NC respectively using riboFECT™ CP Transfection Kit (C10511-05, ribo, Guangzhou, China) according to the manufacturer's instructions. Astrocytes were collected 24 h post transfection, and total cellular protein and mRNA were extracted for western blotting and Real-time PCR, respectively.

**Protein purification**. The HIF-1α cDNA was cloned into the pET-28a(+) vector (NcoI/XhoI) to generate an N-terminal 6×His-tagged fusion protein. The recombinant plasmid was transformed into E. coli BL21(DE3) cells and grown in LB medium with 50 μg/mL kanamycin at 37 °C until OD600 reached 0.5–0.6. Protein expression was induced with 0.2 mM IPTG at 30 °C for 18 h. E. coli BL21(DE3) were harvested, lysed by sonication, and purified using Ni-NTA affinity chromatography. Further purification was performed by size-exclusion chromatography (20 mM Tris-HCl, 150 mM NaCl, pH 8.0). HIF-1α-containing fractions were pooled, concentrated, and stored at −80 °C with 1× PBS.

**Ultraviolet-visible absorption spectroscopy**. Precisely transfer 3 mL of 0.25 μM HIF-1α solution into a 1 cm quartz cuvette. Using PBS as reference, measure its absorption spectrum. Then sequentially add $Pb^{2+}$ solution, mix thoroughly, and incubate at room temperature (298 K) for 1 h until reaction equilibrium is reached. Subsequently measure the absorption spectrum within the 180–300 nm wavelength range using UV/Vis spectrophotometer (Lambda 365; PerkinElmer, USA).

**Fluorescence spectroscopy**. Precisely transfer 3 mL of 0.25 μM HIF-1α solution into a 1 cm quartz cuvette. Place the cuvette in a constant-temperature water bath and equilibrate for 1 h before measuring the emission spectrum and synchronous fluorescence spectrum of HIF-1α. Then sequentially add $Pb^{2+}$ solution, mix thoroughly, and after maintaining at 298 K for 1 h, measure the fluorescence spectrum of the mixed solution using Steady-state fluorescence spectra (FS5; PerkinElmer, USA).

**Circular dichroism spectroscopy**. Precisely pipette 3 mL each of the 0.25 μM HIF-1α or 0.25 μM HIF-1α + 0.5 μM $Pb^{2+}$ solutions into 1 cm quartz cuvettes. Using PBS buffer solution as reference, measure the circular dichroism (CD) spectra of both HIF-1α alone and HIF-1α-$Pb^{2+}$ interaction system. Set the scanning range from 180 to 300 nm using Chirascan circular dichroism spectrophotometer (Applied Photophysics, UK).

## Statistics and reproducibility

All statistical analyses were conducted using the R software (version 4.2.3) for Windows. Normal distributions of data are expressed as the $\bar{x}\pm s$. Normality of variables was assessed with the Shapiro–Wilk test and by visual evaluation of histograms and quantile–quantile plots. Additionally, the equality of variance was examined using the Levene's test. If the data met the assumptions of normality and homogeneity of variance, parametric tests were employed for statistical analysis. When the data do not fit a normal distribution, non-parametric tests are used. One-way ANOVA followed by

Tukey's post hoc test was used for group analyses. Student's $t$-test was used for statistical analyses between the two groups. The significance of enrichment was assessed using a hypergeometric distribution. All correlation analyses were performed using the Pearson's correlation coefficient (r).

## Reporting summary

Further information on research design is available in the Nature Portfolio Reporting Summary linked to this article.

## Data availability

Single-cell RNA-sequence raw data are available in the Sequence Read Archive (SRA) under the accession number PRJNA1310027. The source data behind the graphs in the paper can be found in Supplementary Data. Figures 1a and 2a were created with BioGDP.com. The platform permits the incorporation of its website elements when the article is published[55].

## Code availability

Codes used to perform the analysis are available on GitHub: https://github.com/Biubiufan/manuscript-code.git. The specific computational tools and resources utilized in our single-cell analysis include: Seurat version 4.1.1, Monocle2 version 2.22.0, Augur version 1.0.0.

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

## Acknowledgements

This research was supported by National Natural Science Foundation of China (Grant Number: 82073598) and Science and Technology Project of Hebei Education Department (Grant Number: BJK2024170).

## Author contributions

Conceived and designed experiments: Y.S.Z., and W.X.W., Performed the experiments: F.S., Y.R.W., H.H., Y.W.Z., F.W.W., Y.L.G., S.C.L., and PC.L.; Analyzed the data: F.S., H.H., Wrote the manuscript: F.S., Y.R.W., Revised the article: Y.S.Z., and W.X.W.; All authors approved the final version of the manuscript.

## Competing interests

The authors declare no competing interests.
