## [Transparent Peer Review file · Communications Biology]

Lead binds HIF-1 α contributing to Depression-like Behaviour through Modulating Mitochondria-Associated Astrocyte Ferroptosis

Corresponding Author: Professor Yanshu Zhang

Version 0:

Reviewer comments:

Reviewer #1

(Remarks to the Author)

The paper focuses on the relationship between environmental factors (lead exposure) and depression. Given the high incidence of depression and the prevalence of environmental lead exposure, this research topic has significant public health implications and provides new insights into the pathogenesis of depression. A variety of experimental techniques are comprehensively applied, These techniques are used to conduct in - depth investigations at the whole - animal, cell, and molecular levels, providing multi - dimensional evidence for the research results and enhancing the reliability of the research conclusions. It is revealed for the first time that astrocytes are the most sensitive cell type to ferroptosis in the mouse brain after lead exposure, and HIF - 1 α is an early driver gene of astrocyte ferroptosis. Lead ions inhibit the degradation of HIF - 1 α by directly binding to it, and then regulate VDAC1 - mediated mitochondrial dysfunction and ferroptosis. These findings provide a new molecular mechanism for lead - induced neurotoxicity and depression research. I recommend accepting this manuscript for publication.

Reviewer #2

(Remarks to the Author)

The manuscript "Lead binds HIF-1 α contributing to Depression-like Behaviour through Modulating Mitochondria-Associated Astrocyte Ferroptosis" investigates the relationship between lead (Pb) exposure and depression-like behavior, specifically focusing on ferroptosis in astrocytes as a key mechanistic pathway. The study is in general well conceived, however, the experimental design and interpretation have some limitations that should be addressed to strengthen the manuscript:-

1. Different doses of Pb were applied to test the behavioral effects. How could these doses be compared to human exposure levels?
2. To evaluate depression-like symptoms, the study uses a variety of behavioral tests, including the SPT, FST, TST, OFT, and EPM. Results could be distorted by these assays' susceptibility to outside influences such as handling effects and stress.
3. Additional control groups exposed to other heavy metals should also be kept in the study design to strengthen specificity claims regarding Pb-induced effects.
4. The role of HIF-1 α in regulating VDAC1 and mitochondrial dysfunction is well established, but whether this mechanism is unique to Pb exposure or a general ferroptotic response is unclear. HIF-1 also regulates apoptosis by promoting the formation of VDAC1- Δ C from VDAC1. Given the overlap between ferroptosis and other cell death mechanisms, how were necroptosis and apoptosis ruled out as primary mechanisms of Pb-induced astrocyte death?
5. Were any biochemical methods, such as Surface Plasmon Resonance, used to confirm Pb-HIF-1 α binding experimentally?
6. The study measures depression-like behavior in mice following 8 weeks of Pb exposure. It is suggested to include longer-term studies to assess the persistent effects of Pb.

Besides addressing these issues the authors must revise thoroughly to eliminate grammatical and spelling errors.

Reviewer #3

(Remarks to the Author)

In this manuscript, the authors investigate whether ferroptosis is involved in depression-like behavior in a Pb-exposed mouse model. Furthermore, they identify the key regulators of the proposed Pb mechanism, namely HIF-1 alpha and VDAC. The manuscript is well-written, clear, and easy to follow. Statistical analysis is appropriate, and the detailed materials and methods section allows researchers to reproduce the work. However, I have some concerns, as follows:

Major concerns:

- i. It is not immediately clear that “target protein relative expression” refers to WB band quantification. I suggest using “target protein quantification.”
- ii. It is not entirely obvious why Fer-1, a ferroptosis inhibitor, has no effect on animal behavior tests. This should be discussed in the Discussion section.
- iii. The authors present Pb protein content in the cytoplasmic and nuclear fractions. Once activated, HIF-1 α translocates to the nucleus, leading to a decrease in its cytoplasmic levels. The authors state that “a higher level of HIF-1 α protein expression was observed in the nucleus”; however, there is no Figure showing statistical significance between the cytoplasmic and nuclear fractions. This distinction is not evident in Figure 5D.
- iv. It is quite unusual that siRNA HIF-1 α and siRNA VDAC had no effect in any of the experiments conducted, with effects only observed after Pb exposure. It would be important to show transfection efficiency, including in the presence of Pb.
- v. Considering the effect of VDAC after Pb exposure, did you measure mitochondrial bioenergetics under these experimental conditions? It would be interesting to observe its effects.

Minor concerns:

- i) In Figures 3A and 3B, the colors should be consistent for the same cell subtype, as the differing colors make it difficult to associate the two graphs.
- ii) Some spaces are missing both in the article and in the captions of the supplementary figures.
- iii) The terms *in vitro* and *in vivo* should be italicized (page 24).

Version 1:

Reviewer comments:

Reviewer #1

(Remarks to the Author)

The paper focuses on the relationship between environmental factors (lead exposure) and depression. Given the high incidence of depression and the prevalence of environmental lead exposure, this research topic has significant public health implications and provides new insights into the pathogenesis of depression. A variety of experimental techniques are comprehensively applied. These techniques are used to conduct in - depth investigations at the whole - animal, cell, and molecular levels, providing multi - dimensional evidence for the research results and enhancing the reliability of the research conclusions. It is revealed for the first time that astrocytes are the most sensitive cell type to ferroptosis in the mouse brain after lead exposure, and HIF-1 α is an early driver gene of astrocyte ferroptosis. Lead ions inhibit the degradation of HIF - 1 α by directly binding to it, and then regulate VDAC1- mediated mitochondrial dysfunction and ferroptosis. These findings provide a new molecular mechanism for lead-induced neurotoxicity and depression research. I recommend accepting this manuscript for publication.

Reviewer #2

(Remarks to the Author)

As the authors have now addressed all the issues raised, I recommend acceptance of the manuscript.

Reviewer #3

(Remarks to the Author)

The authors have clearly addressed all my concerns. I support the publication of the revised version of the manuscript.

Response to the Reviewer #1:

The paper focuses on the relationship between environmental factors (lead exposure) and depression. Given the high incidence of depression and the prevalence of environmental lead exposure, this research topic has significant public health implications and provides new insights into the pathogenesis of depression. A variety of experimental techniques are comprehensively applied. These techniques are used to conduct in - depth investigations at the whole - animal, cell, and molecular levels, providing multi - dimensional evidence for the research results and enhancing the reliability of the research conclusions. It is revealed for the first time that astrocytes are the most sensitive cell type to ferroptosis in the mouse brain after lead exposure, and HIF-1 α is an early driver gene of astrocyte ferroptosis. Lead ions inhibit the degradation of HIF - 1 α by directly binding to it, and then regulate VDAC1- mediated mitochondrial dysfunction and ferroptosis. These findings provide a new molecular mechanism for lead-induced neurotoxicity and depression research. I recommend accepting this manuscript for publication.

Response: We sincerely appreciate your insightful comments and positive evaluation of our work. This constructive recommendation is particularly encouraging, providing clear guidance for determining our subsequent research direction.

Response to the Reviewer #2:

The manuscript "Lead binds HIF-1 α contributing to Depression-like Behaviour through Modulating Mitochondria-Associated Astrocyte Ferroptosis" investigates the relationship between lead (Pb) exposure and depression-like behavior, specifically focusing on ferroptosis in astrocytes as a key mechanistic pathway. The study is in general well conceived, however, the experimental design and interpretation have some limitations that should be addressed to strengthen the manuscript:

1. Different doses of Pb were applied to test the behavioral effects. How could these doses be compared to human exposure levels?

Response: To simulate human-relevant exposure levels, we administered mice with 12.5, 25, and 50 mg/kg PbAc via gavage for 8 weeks, resulting in blood Pb concentrations of 69.23 $\mu\text{g/L}$ (LPb), 127.93 $\mu\text{g/L}$ (MPb), and 208.23 $\mu\text{g/L}$ (HPb),

respectively. The blood Pb in LPb and MPb groups fell within the range of China's reference blood lead level¹, while blood Pb in HPb concentrations approached those observed in occupationally exposed populations. Meanwhile, the current dosage is similar to Wang. et al, Xie. et al, Zhou. et al, Li. et al study, which commonly used to research in term of neurotoxicity induced by environmental or occupationally lead exposure²⁻⁵.

1. Lyu, Y. et al. Declines in blood lead levels among general population - china, 2000-2018. *China CDC Weekly* **4**, 1117-1122 (2022).
2. Wang, N. et al. Lead exposure exacerbates liver injury in high-fat diet-fed mice by disrupting the gut microbiota and related metabolites. *Food Funct.* **15**, 3060-3075 (2024).
3. Xie, X. et al. Potential mechanisms of aortic medial degeneration promoted by co-exposure to microplastics and lead. *J. Hazard. Mater.* **475**, 134854 (2024).
4. Zhou, R. et al. Chronic pb exposure induces anxiety and depression-like behaviors in mice via excitatory neuronal hyperexcitability in ventral hippocampal dentate gyrus. *Environ. Sci. Technol.* **57**, 12222-12233 (2023).
5. Li, Y. et al. Sodium butyrate alleviates lead-induced neuroinflammation and improves cognitive and memory impairment through the acss2/h3k9ac/bdnf pathway. *Environ. Int.* **184**, 108479 (2024).

2. To evaluate depression-like symptoms, the study uses a variety of behavioral tests, including the SPT, FST, TST, OFT, and EPM. Results could be distorted by these assays' susceptibility to outside influences such as handling effects and stress.

Response: We sincerely appreciate your insightful comments regarding potential confounding factors in behavioral tests. To minimize interference from outside influences, all behavioral tests were conducted during the same circadian period (9:00-11:00 AM). Mice were acclimated to experimenters and handling procedures for 3 days prior to testing. An automated video tracking system was employed for data recording to minimize human intervention. These standardized protocols significantly reduced the confounding factors you mentioned." In subsequent studies, we will optimize the

behavioral testing.

3. Additional control groups exposed to other heavy metals should also be kept in the study design to strengthen specificity claims regarding Pb-induced effects.

Response: Thank you for your suggestion. As you rightly pointed out, the current data cannot conclusively establish ferroptosis as a Pb-specific mechanism of neurotoxicity. Other studies demonstrated that other heavy metals like Fe^{3+} and Mn^{2+} can also induce ferroptosis in brain tissues^{6,7}. The primary objective of this study was to elucidate the mechanistic role of ferroptosis in Pb exposure-induced depression-like behaviors, which explains the absence of additional heavy metal control groups in our experimental design. We have explicitly acknowledged this limitation in the revised Discussion section (revised manuscript, Page 45 line 28- Page 46 line 3).

To strengthen specificity claims regarding Pb-induced effects, we conducted computational docking analyses of several metal ions (Fe^{3+} , Mn^{2+} , Zn^{2+}) with the Pro138 site of HIF-1 α as an indirect approach. Our results suggest that Pb^{2+} exhibits relatively stronger binding affinity ($\Delta G = -2.13$ kcal/mol) than the other tested ions (Fe^{3+} : -1.89 kcal/mol; Mn^{2+} : -1.97 kcal/mol; Zn^{2+} : -1.65 kcal/mol), indicating a potential preferential interaction (Fig. S7A) (revised manuscript, Page 32 line 22- 26).

Figure S7 A: Molecular docking model illustrating the binding of metal to HIF-1 α and the SDS-polyacrylamide gel preparation for HIF-1 α protein.

6. Shi, W. et al. Co-exposure to fe, zn, and cu induced neuronal ferroptosis with associated lipid metabolism disorder via the erk/cpla2/aa pathway. *Environ. Pollut.* **336**, 122438 (2023).
7. Zhang, S. et al. Multi-omics analysis reveals mn exposure affects ferroptosis pathway in zebrafish brain. *Ecotoxicol. Environ. Saf.* **253**, 114616 (2023).

4. The role of HIF-1 α in regulating VDAC1 and mitochondrial dysfunction is well established, but whether this mechanism is unique to Pb exposure or a general ferroptotic response is unclear. HIF-1 also regulates apoptosis by promoting the

formation of VDAC1- Δ C from VDAC1. Given the overlap between ferroptosis and other cell death mechanisms, how were necroptosis and apoptosis ruled out as primary mechanisms of Pb-induced astrocyte death?

Response: We appreciate your insightful questions regarding cell death mechanisms. In our initial experiments, we employed necrostatin-1 (Nec), Z-VAD-FMK (Z-VAD), and ferrostatin-1 (Fer-1) to treat Pb-exposed astrocytes. The results demonstrated that both Z-VAD and Fer-1 significantly attenuated the Pb-induced decreasing of astrocyte viability, whereas Nec showed no protective effect. To further investigate the relative contributions of ferroptosis and apoptosis in Pb-induced astrocyte death, we examined the expression of apoptosis-related proteins. The results demonstrated that 5 μ M Pb exposure did not alter the expression of the apoptosis-related protein caspase-3 in astrocytes. Notably, at 10 μ M and 20 μ M Pb exposure, the fold changes in terms of apoptotic protein expression were substantially lower than those observed the ferroptosis marker PTGS2. Collectively, these findings suggest that ferroptosis plays a more dominant role than apoptosis or necroptosis in Pb-induced astrocyte death.

Figure S5 D: Gel electrophoresis plot and semi-quantification of Caspase-3 in different group.

** $P < 0.01$ vs Control; *** $P < 0.001$ vs Control. Comparisons are made with ANOVA, followed by Tukey's multiple comparison tests.

5. Were any biochemical methods, such as Surface Plasmon Resonance, used to confirm Pb-HIF-1 α binding experimentally?

Response: Thank you for your suggestion. After evaluating the Surface Plasmon Resonance (SPR) methodology, we found that this technique is not an optimal method for analyzing metal-protein binding interactions, which primarily detects interactions

with molecular weights >200 Da (Pb^{2+} : 207 Da, near the detection threshold)⁸. Thus, we employed ultraviolet spectroscopy, fluorescence spectroscopy, and circular dichroism instead of surface plasmon resonance to characterize Pb^{2+} -HIF-1 α interactions. These methods are commonly employed to investigate metal-protein binding interactions^{9,10}. The results of ultraviolet spectroscopy showed that as the concentration of Pb^{2+} increased, the intensity of the ligand-to-metal charge transfer band progressively intensified, exhibiting a dominant peak at 225 nm and a shoulder peak at 205 nm (Fig.6 J). Fluorescence spectroscopy revealed that the fluorescence intensity of the Pb-HIF-1 α complex increased with increasing concentrations of Pb^{2+} , indicating an interaction between Pb^{2+} and HIF-1 α (Fig.6 K). Additionally, upon the addition of Pb^{2+} , the negative peak intensity in the CD spectrum of HIF-1 α increased suggesting a rise in α -helical content. This structural alteration of HIF-1 α likely arises from the binding of positively charged Pb^{2+} to negatively charged residues in HIF-1 α , which modifies its secondary structure via non-covalent interactions and enhances protein stability by facilitating α -helix formation (Fig.6 L).

Figure 6: **J.** UV-Vis absorption spectra, **K.** Synchronous fluorescence spectra and **L.** Circular dichroism spectrum of HIF-1 α and Pb^{2+} - HIF-1 α systems (T = 298 K), C(HIF-1 α)= 0.25 μM , C(Pb^{2+}) (1 \rightarrow 4): 0, 0.5, 1, 2 μM .

8. Camarca, A. et al. Emergent biosensing technologies based on fluorescence spectroscopy and surface plasmon resonance. *Sensors* **21**, (2021).
9. Berntsson, E. et al. Binding of hg(i) and hg(ii) ions to amyloid-beta (abeta) peptide variants affect their structure and aggregation. *ChemBioChem* e202500252 (2025).
10. Chunmei, D. et al. Study of the interaction between mercury (ii) and bovine serum albumin by spectroscopic methods. *Environ. Toxicol. Pharmacol.* **37**, 870-877

(2014).

6. The study measures depression-like behavior in mice following 8 weeks of Pb exposure. It is suggested to include longer-term studies to assess the persistent effects of Pb.

Response: Thank you for your suggestion. In our ongoing unpublished research, we have systematically investigated the temporal progression of Pb-induced neurobehavioral alterations. To assess the long-term effects of Pb exposure induced depression-like behaviors in mice, we examined the immobility time of FST and sucrose preference for 12 and 24 w. The results demonstrated that Pb exposure induced a sustained reduction in sucrose preference rate in mice up to 24 w, whereas the immobility time in the FST peaked at 12 w (Fig S2 B, C).

Figure S2 B, C: B. The sucrose preference and C. immobility time of FST in Control, Pb group.

7. Besides addressing these issues the authors must revise thoroughly to eliminate grammatical and spelling errors.

Response: Thank you for your suggestion. We have thoroughly reviewed the entire manuscript and corrected all grammatical and spelling errors.

Response to the Reviewer #3:

In this manuscript, the authors investigate whether ferroptosis is involved in depression-like behavior in a Pb-exposed mouse model. Furthermore, they identify the key regulators of the proposed Pb mechanism, namely HIF-1 alpha and VDAC. The manuscript is well-written, clear, and easy to follow. Statistical analysis is appropriate, and the detailed materials and methods section allows researchers to reproduce the work.

However, I have some concerns, as follows:

Major concerns:

1. It is not immediately clear that “target protein relative expression” refers to WB band quantification. I suggest using “target protein quantification.”

Response: We appreciate your suggestion. Accordingly, we have revised "target protein relative expression" to "target protein quantification in all figures.

2. It is not entirely obvious why Fer-1, a ferroptosis inhibitor, has no effect on animal behavior tests. This should be discussed in the Discussion section.

Response: Thank you for your suggestion. As suggested, we have expanded the discussion regarding of the lack of behavioral effects observed with Fer-1 treatment (Page 44, Lines 1–12 in the revised manuscript). The observed lack of behavioral changes may be attributed to the administered dosage of Fer-1. In present study, 2 mg/kg Fer-1 was used to animal experiment, which were based on the manufacturer's specifications and relevant literature (the recommended dosage range for Fer-1 in animal studies is 0.8-10 mg/kg). Wu et al study applied 5 mg/kg Fer-1 with every fourth day for 28 days to explore the Nitrogen-doped graphene quantum dots induced the learning and memory impairment. Sripetchwandee et al study employed 2 mg/kg Fer-1 to explore the iron-overloaded induced cognitive disorder. Notably, none of the doses within this range have been reported to induce behavioral changes in mice, which is consistent with our findings^{11,12}.

11. Wu, T. et al. Induction of ferroptosis in response to graphene quantum dots through mitochondrial oxidative stress in microglia. *Part. Fibre Toxicol.* **17**, 30 (2020).

12. Sripetchwandee, J. et al. Ferrostatin-1 and z-vad-fmk potentially attenuated iron-mediated neurotoxicity and rescued cognitive function in iron-overloaded rats. *Life Sci.* **313**, 121269 (2023).

3. The authors present Pb protein content in the cytoplasmic and nuclear fractions. Once activated, HIF-1 α translocates to the nucleus, leading to a decrease in its cytoplasmic levels. The authors state that “a higher level of HIF-1 α protein expression was observed in the nucleus”; however, there is no Figure showing statistical significance between

the cytoplasmic and nuclear fractions. This distinction is not evident in Figure 5D.

Response: We sincerely appreciate this insightful observation regarding the subcellular distribution of HIF-1 α . Upon re-examining our data, we found that Pb exposure increased HIF-1 α protein expression in both nuclear and cytoplasmic fractions. This dual increase possible suggests that Pb exposure upregulates *Hif-1 α* mRNA expression, thereby enhancing overall protein translation. Although HIF-1 α undergoes nuclear translocation, its upregulated synthesis maintains detectable levels in the cytoplasm following Pb exposure in astrocyte, which were similar with Liang. et al, Wu. et al study^{13,14}. To emphasize the statistical significance between the cytoplasmic and nuclear fractions, we quantified the fold change of HIF-1 α protein in both cytoplasmic and nuclear fractions relative to the control group on the Results section (revised manuscript, Page 28 line 12-18).

13. Liang, Z. et al. The role of hif-1alpha/ho-1 pathway in hippocampal neuronal ferroptosis in epilepsy. *iScience* **26**, 108098 (2023).

14. Wu, Y. et al. Di-(2-ethylhexyl) phthalate exposure leads to ferroptosis via the hif-1alpha/ho-1 signaling pathway in mouse testes. *J. Hazard. Mater.* **426**, 127807 (2022).

4. It is quite unusual that siRNA HIF-1 α and siRNA VDAC had no effect in any of the experiments conducted, with effects only observed after Pb exposure. It would be important to show transfection efficiency, including in the presence of Pb.

Response: Thank you for your suggestion. Our transfection efficiency was showed in Figure S6 and S8. Knockdown of either *Hif-1 α* or *Vdac1* significantly reduced the protein expression levels of HIF-1 α (to 52% of control values) or VDAC1 (to 54% of control) in astrocytes, respectively. Indeed, significant differences were observed in Fe²⁺ content, Percent of JC-1, GSH content between the si-HIF-1 α or si-VDAC1 groups compared to the control group. The statistical significance has been explicitly annotated in the corresponding figures including Fig 5E, Fig 8 G, Fig 9D.

5. Considering the effect of VDAC after Pb exposure, did you measure mitochondrial bioenergetics under these experimental conditions? It would be interesting to observe

its effects.

Response: Thank you for your suggestion. We fully acknowledge the importance of measuring mitochondrial bioenergetics in our study. Because of the absence of a Seahorse XF analyzer for measuring mitochondrial bioenergetics in our university. Alternative indicators were employed to assess mitochondrial bioenergetics, including cellular ATP content and superoxide (SOX) levels. Our results showed that decreased ATP content and elevated SOX observed in astrocyte following Pb exposure, which indicate that mitochondrial bioenergetic dysfunction.

Minor concerns:

1. In Figures 3A and 3B, the colors should be consistent for the same cell subtype, as the differing colors make it difficult to associate the two graphs.

Response: Thank you for your suggestion. We used the same color in same cell subtype of Figures 3A and 3B.

2. Some spaces are missing both in the article and in the captions of the supplementary figures.

Response: Thank you for your suggestion. We have inserted spaces in the article and in the captions of the supplementary figures.

3. The terms *in vitro* and *in vivo* should be italicized (page 24).

Response: Thank you for your suggestion. The terms "*in vitro*" and "*in vivo*" on page 24 have been italicized and highlighted in red.

Response to the Reviewer #1:

The paper focuses on the relationship between environmental factors (lead exposure) and depression. Given the high incidence of depression and the prevalence of environmental lead exposure, this research topic has significant public health implications and provides new insights into the pathogenesis of depression. A variety of experimental techniques are comprehensively applied. These techniques are used to conduct in - depth investigations at the whole - animal, cell, and molecular levels, providing multi - dimensional evidence for the research results and enhancing the reliability of the research conclusions. It is revealed for the first time that astrocytes are the most sensitive cell type to ferroptosis in the mouse brain after lead exposure, and HIF-1 α is an early driver gene of astrocyte ferroptosis. Lead ions inhibit the degradation of HIF - 1 α by directly binding to it, and then regulate VDAC1- mediated mitochondrial dysfunction and ferroptosis. These findings provide a new molecular mechanism for lead-induced neurotoxicity and depression research. I recommend accepting this manuscript for publication.

Response: We sincerely appreciate your insightful comments and positive evaluation of our work. This constructive recommendation is particularly encouraging, providing clear guidance for determining our subsequent research direction.

Response to the Reviewer #2:

The manuscript "Lead binds HIF-1 α contributing to Depression-like Behaviour through Modulating Mitochondria-Associated Astrocyte Ferroptosis" investigates the relationship between lead (Pb) exposure and depression-like behavior, specifically focusing on ferroptosis in astrocytes as a key mechanistic pathway. The study is in general well conceived, however, the experimental design and interpretation have some limitations that should be addressed to strengthen the manuscript:

1. Different doses of Pb were applied to test the behavioral effects. How could these doses be compared to human exposure levels?

Response: To simulate human-relevant exposure levels, we administered mice with 12.5, 25, and 50 mg/kg PbAc via gavage for 8 weeks, resulting in blood Pb concentrations of 69.23 $\mu\text{g/L}$ (LPb), 127.93 $\mu\text{g/L}$ (MPb), and 208.23 $\mu\text{g/L}$ (HPb),

respectively. The blood Pb in LPb and MPb groups fell within the range of China's reference blood lead level¹, while blood Pb in HPb concentrations approached those observed in occupationally exposed populations. Meanwhile, the current dosage is similar to Wang. et al, Xie. et al, Zhou. et al, Li. et al study, which commonly used to research in term of neurotoxicity induced by environmental or occupationally lead exposure²⁻⁵.

1. Lyu, Y. et al. Declines in blood lead levels among general population - china, 2000-2018. *China CDC Weekly* **4**, 1117-1122 (2022).
2. Wang, N. et al. Lead exposure exacerbates liver injury in high-fat diet-fed mice by disrupting the gut microbiota and related metabolites. *Food Funct.* **15**, 3060-3075 (2024).
3. Xie, X. et al. Potential mechanisms of aortic medial degeneration promoted by co-exposure to microplastics and lead. *J. Hazard. Mater.* **475**, 134854 (2024).
4. Zhou, R. et al. Chronic pb exposure induces anxiety and depression-like behaviors in mice via excitatory neuronal hyperexcitability in ventral hippocampal dentate gyrus. *Environ. Sci. Technol.* **57**, 12222-12233 (2023).
5. Li, Y. et al. Sodium butyrate alleviates lead-induced neuroinflammation and improves cognitive and memory impairment through the acss2/h3k9ac/bdnf pathway. *Environ. Int.* **184**, 108479 (2024).

2. To evaluate depression-like symptoms, the study uses a variety of behavioral tests, including the SPT, FST, TST, OFT, and EPM. Results could be distorted by these assays' susceptibility to outside influences such as handling effects and stress.

Response: We sincerely appreciate your insightful comments regarding potential confounding factors in behavioral tests. To minimize interference from outside influences, all behavioral tests were conducted during the same circadian period (9:00-11:00 AM). Mice were acclimated to experimenters and handling procedures for 3 days prior to testing. An automated video tracking system was employed for data recording to minimize human intervention. These standardized protocols significantly reduced the confounding factors you mentioned." In subsequent studies, we will optimize the

behavioral testing.

3. Additional control groups exposed to other heavy metals should also be kept in the study design to strengthen specificity claims regarding Pb-induced effects.

Response: Thank you for your suggestion. As you rightly pointed out, the current data cannot conclusively establish ferroptosis as a Pb-specific mechanism of neurotoxicity. Other studies demonstrated that other heavy metals like Fe^{3+} and Mn^{2+} can also induce ferroptosis in brain tissues^{6,7}. The primary objective of this study was to elucidate the mechanistic role of ferroptosis in Pb exposure-induced depression-like behaviors, which explains the absence of additional heavy metal control groups in our experimental design. We have explicitly acknowledged this limitation in the revised Discussion section (revised manuscript, Page 43 line 28- Page 44 line 3).

To strengthen specificity claims regarding Pb-induced effects, we conducted computational docking analyses of several metal ions (Fe^{3+} , Mn^{2+} , Zn^{2+}) with the Pro138 site of HIF-1 α as an indirect approach. Our results suggest that Pb^{2+} exhibits relatively stronger binding affinity ($\Delta G = -2.13$ kcal/mol) than the other tested ions (Fe^{3+} : -1.89 kcal/mol; Mn^{2+} : -1.97 kcal/mol; Zn^{2+} : -1.65 kcal/mol), indicating a potential preferential interaction (Fig. S7A).

Figure S7 A: Molecular docking model illustrating the binding of metal to HIF-1 α and the SDS-polyacrylamide gel preparation for HIF-1 α protein.

6. Shi, W. et al. Co-exposure to fe, zn, and cu induced neuronal ferroptosis with associated lipid metabolism disorder via the erk/cpla2/aa pathway. *Environ. Pollut.* **336**, 122438 (2023).
7. Zhang, S. et al. Multi-omics analysis reveals mn exposure affects ferroptosis pathway in zebrafish brain. *Ecotoxicol. Environ. Saf.* **253**, 114616 (2023).

4. The role of HIF-1 α in regulating VDAC1 and mitochondrial dysfunction is well established, but whether this mechanism is unique to Pb exposure or a general ferroptotic response is unclear. HIF-1 also regulates apoptosis by promoting the

formation of VDAC1- Δ C from VDAC1. Given the overlap between ferroptosis and other cell death mechanisms, how were necroptosis and apoptosis ruled out as primary mechanisms of Pb-induced astrocyte death?

Response: We appreciate your insightful questions regarding cell death mechanisms. In our initial experiments, we employed necrostatin-1 (Nec), Z-VAD-FMK (Z-VAD), and ferrostatin-1 (Fer-1) to treat Pb-exposed astrocytes. The results demonstrated that both Z-VAD and Fer-1 significantly attenuated the Pb-induced decreasing of astrocyte viability, whereas Nec showed no protective effect. To further investigate the relative contributions of ferroptosis and apoptosis in Pb-induced astrocyte death, we examined the expression of apoptosis-related proteins. The results demonstrated that 5 μ M Pb exposure did not alter the expression of the apoptosis-related protein caspase-3 in astrocytes. Notably, at 10 μ M and 20 μ M Pb exposure, the fold changes in terms of apoptotic protein expression were substantially lower than those observed the ferroptosis marker PTGS2. Collectively, these findings suggest that ferroptosis plays a more dominant role than apoptosis or necroptosis in Pb-induced astrocyte death.

Figure S5 D: Gel electrophoresis plot and semi-quantification of Caspase-3 in different group.

** $P < 0.01$ vs Control; *** $P < 0.001$ vs Control. Comparisons are made with ANOVA, followed by Tukey's multiple comparison tests.

5. Were any biochemical methods, such as Surface Plasmon Resonance, used to confirm Pb-HIF-1 α binding experimentally?

Response: Thank you for your suggestion. After evaluating the Surface Plasmon Resonance (SPR) methodology, we found that this technique is not an optimal method for analyzing metal-protein binding interactions, which primarily detects interactions

with molecular weights >200 Da (Pb^{2+} : 207 Da, near the detection threshold)⁸. Thus, we employed ultraviolet spectroscopy, fluorescence spectroscopy, and circular dichroism instead of surface plasmon resonance to characterize Pb^{2+} -HIF-1 α interactions. These methods are commonly employed to investigate metal-protein binding interactions^{9,10}. The results of ultraviolet spectroscopy showed that as the concentration of Pb^{2+} increased, the intensity of the ligand-to-metal charge transfer band progressively intensified, exhibiting a dominant peak at 225 nm and a shoulder peak at 205 nm (Fig.6 J). Fluorescence spectroscopy revealed that the fluorescence intensity of the Pb -HIF-1 α complex increased with increasing concentrations of Pb^{2+} , indicating an interaction between Pb^{2+} and HIF-1 α (Fig.6 K). Additionally, upon the addition of Pb^{2+} , the negative peak intensity in the CD spectrum of HIF-1 α increased suggesting a rise in α -helical content. This structural alteration of HIF-1 α likely arises from the binding of positively charged Pb^{2+} to negatively charged residues in HIF-1 α , which modifies its secondary structure via non-covalent interactions and enhances protein stability by facilitating α -helix formation (Fig.6 L).

Figure 6: **J.** UV-Vis absorption spectra, **K.** Synchronous fluorescence spectra and **L.** Circular dichroism spectrum of HIF-1 α and Pb^{2+} - HIF-1 α systems (T = 298 K), C(HIF-1 α)= 0.25 μM , C(Pb^{2+}) (1→4): 0, 0.5, 1, 2 μM .

8. Camarca, A. et al. Emergent biosensing technologies based on fluorescence spectroscopy and surface plasmon resonance. *Sensors* **21**, (2021).
9. Berntsson, E. et al. Binding of hg(i) and hg(ii) ions to amyloid-beta (abeta) peptide variants affect their structure and aggregation. *ChemBioChem* e202500252 (2025).
10. Chunmei, D. et al. Study of the interaction between mercury (ii) and bovine serum albumin by spectroscopic methods. *Environ. Toxicol. Pharmacol.* **37**, 870-877

(2014).

6. The study measures depression-like behavior in mice following 8 weeks of Pb exposure. It is suggested to include longer-term studies to assess the persistent effects of Pb.

Response: Thank you for your suggestion. In our ongoing unpublished research, we have systematically investigated the temporal progression of Pb-induced neurobehavioral alterations. To assess the long-term effects of Pb exposure induced depression-like behaviors in mice, we examined the immobility time of FST and sucrose preference for 12 and 24 w. The results demonstrated that Pb exposure induced a sustained reduction in sucrose preference rate in mice up to 24 w, whereas the immobility time in the FST peaked at 12 w (Fig S2 B, C).

Figure S2 B, C: B. The sucrose preference and C. immobility time of FST in Control, Pb group.

7. Besides addressing these issues the authors must revise thoroughly to eliminate grammatical and spelling errors.

Response: Thank you for your suggestion. We have thoroughly reviewed the entire manuscript and corrected all grammatical and spelling errors.

Response to the Reviewer #3:

In this manuscript, the authors investigate whether ferroptosis is involved in depression-like behavior in a Pb-exposed mouse model. Furthermore, they identify the key regulators of the proposed Pb mechanism, namely HIF-1 alpha and VDAC. The manuscript is well-written, clear, and easy to follow. Statistical analysis is appropriate, and the detailed materials and methods section allows researchers to reproduce the work.

However, I have some concerns, as follows:

Major concerns:

1. It is not immediately clear that “target protein relative expression” refers to WB band quantification. I suggest using “target protein quantification.”

Response: We appreciate your suggestion. Accordingly, we have revised "target protein relative expression" to "target protein quantification in all figures.

2. It is not entirely obvious why Fer-1, a ferroptosis inhibitor, has no effect on animal behavior tests. This should be discussed in the Discussion section.

Response: Thank you for your suggestion. As suggested, we have expanded the discussion regarding of the lack of behavioral effects observed with Fer-1 treatment (Page 42, Lines 4–11 in the revised manuscript). The observed lack of behavioral changes may be attributed to the administered dosage of Fer-1. In present study, 2 mg/kg Fer-1 was used to animal experiment, which were based on the manufacturer's specifications and relevant literature (the recommended dosage range for Fer-1 in animal studies is 0.8-10 mg/kg). Wu et al study applied 5 mg/kg Fer-1 with every fourth day for 28 days to explore the Nitrogen-doped graphene quantum dots induced the learning and memory impairment. Sripetchwandee et al study employed 2 mg/kg Fer-1 to explore the iron-overloaded induced cognitive disorder. Notably, none of the doses within this range have been reported to induce behavioral changes in mice, which is consistent with our findings^{11,12}.

11. Wu, T. et al. Induction of ferroptosis in response to graphene quantum dots through mitochondrial oxidative stress in microglia. *Part. Fibre Toxicol.* **17**, 30 (2020).

12. Sripetchwandee, J. et al. Ferrostatin-1 and z-vad-fmk potentially attenuated iron-mediated neurotoxicity and rescued cognitive function in iron-overloaded rats. *Life Sci.* **313**, 121269 (2023).

3. The authors present Pb protein content in the cytoplasmic and nuclear fractions. Once activated, HIF-1 α translocates to the nucleus, leading to a decrease in its cytoplasmic levels. The authors state that “a higher level of HIF-1 α protein expression was observed in the nucleus”; however, there is no Figure showing statistical significance between

the cytoplasmic and nuclear fractions. This distinction is not evident in Figure 5D.

Response: We sincerely appreciate this insightful observation regarding the subcellular distribution of HIF-1 α . Upon re-examining our data, we found that Pb exposure increased HIF-1 α protein expression in both nuclear and cytoplasmic fractions. This dual increase possible suggests that Pb exposure upregulates *Hif-1 α* mRNA expression, thereby enhancing overall protein translation. Although HIF-1 α undergoes nuclear translocation, its upregulated synthesis maintains detectable levels in the cytoplasm following Pb exposure in astrocyte, which were similar with Liang. et al, Wu. et al study^{13,14}. To emphasize the statistical significance between the cytoplasmic and nuclear fractions, we quantified the fold change of HIF-1 α protein in both cytoplasmic and nuclear fractions relative to the control group on the Results section (revised manuscript, Page 26 line 12-18).

13. Liang, Z. et al. The role of hif-1alpha/ho-1 pathway in hippocampal neuronal ferroptosis in epilepsy. *iScience* **26**, 108098 (2023).

14. Wu, Y. et al. Di-(2-ethylhexyl) phthalate exposure leads to ferroptosis via the hif-1alpha/ho-1 signaling pathway in mouse testes. *J. Hazard. Mater.* **426**, 127807 (2022).

4. It is quite unusual that siRNA HIF-1 α and siRNA VDAC had no effect in any of the experiments conducted, with effects only observed after Pb exposure. It would be important to show transfection efficiency, including in the presence of Pb.

Response: Thank you for your suggestion. Our transfection efficiency was showed in Figure S6 and S8. Knockdown of either *Hif-1 α* or *Vdac1* significantly reduced the protein expression levels of HIF-1 α (to 52% of control values) or VDAC1 (to 54% of control) in astrocytes, respectively. Indeed, significant differences were observed in Fe²⁺ content, Percent of JC-1, GSH content between the si-HIF-1 α or si-VDAC1 groups compared to the control group. The statistical significance has been explicitly annotated in the corresponding figures including Fig 5E, Fig 8 G, Fig 9D.

5. Considering the effect of VDAC after Pb exposure, did you measure mitochondrial bioenergetics under these experimental conditions? It would be interesting to observe

its effects.

Response: Thank you for your suggestion. We fully acknowledge the importance of measuring mitochondrial bioenergetics in our study. Because of the absence of a Seahorse XF analyzer for measuring mitochondrial bioenergetics in our university. Alternative indicators were employed to assess mitochondrial bioenergetics, including cellular ATP content and superoxide (SOX) levels. Our results showed that decreased ATP content and elevated SOX observed in astrocyte following Pb exposure, which indicate that mitochondrial bioenergetic dysfunction.

Minor concerns:

1. In Figures 3A and 3B, the colors should be consistent for the same cell subtype, as the differing colors make it difficult to associate the two graphs.

Response: Thank you for your suggestion. We used the same color in same cell subtype of Figures 3A and 3B.

2. Some spaces are missing both in the article and in the captions of the supplementary figures.

Response: Thank you for your suggestion. We have inserted spaces in the article and in the captions of the supplementary figures.

3. The terms *in vitro* and *in vivo* should be italicized (page 24).

Response: Thank you for your suggestion. The terms "*in vitro*" and "*in vivo*" on page 24 have been italicized and highlighted in red.